

# Classical weight-four L-value ratios as sums of Calabi-Yau invariants

**Philip Candelas[1⋆], Xenia de la Ossa[1†] and Joseph McGovern[2‡]**

**1** Mathematical Institute, University of Oxford, Radcliffe Observatory Quarter,
Oxford, OX2 6GG, United Kingdom
**2** School of Mathematics and Statistics, University of Melbourne,
Parkville, VIC 3010, Australia

⋆ candelas@maths.ox.ac.uk , † delaossa@maths.ox.ac.uk , ‡ mcgovernjv@gmail.com

## Abstract

We revisit the series solutions of the attractor equations of 4d $\mathcal{N} = 2$ supergravities obtained by Calabi–Yau compactifications previously studied in [1]. While only convergent for a restricted set of black hole charges, we find that they are summable with Padé resummation providing a suitable method. By specialising these solutions to rank-two attractors, we obtain many conjectural identities of the type discovered in [1]. These equate ratios of weight-four special L-values with an infinite series whose summands are formed out of genus-0 Gromov–Witten invariants. We also present two new rank-two attractors which belong to moduli spaces each interesting in their own right. Each of these moduli spaces possess two points of maximal unipotent monodromy. One has already been studied by Hosono and Takagi, and we discuss issues stemming from the associated L-function having nonzero rank.

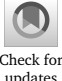

# 1  Introduction

Consider the following, conjectured, identity:

$$\frac{3\pi}{2}\frac{L_{\mathbf{54.4.a.c}}(1)}{L_{\mathbf{54.4.a.c}}(2)} = \sqrt{69} - \sqrt{\frac{2}{\pi^3}}\sum_{\substack{j\in\mathbf{Z}_{>0}\\ \mathfrak{p}\in\mathrm{pt}(j)}}(-1)^j\widetilde{N}_{\mathfrak{p}}\left(\frac{j}{3\pi\sqrt{69}}\right)^{l(\mathfrak{p})-1/2}K_{l(\mathfrak{p})-1/2}\left(\frac{\pi j\sqrt{69}}{3}\right). \quad (1)$$

On the left hand side, we have a simple rational function of special L-values. These quantities have a twofold existence. On the one hand, they are numbers obtained by evaluating the Mellin transform of a weight-four modular form at the integers. On the other hand, these L-values can be calculated from the numbers of points of a manifold, considered as a variety over the finite fields $\mathbf{F}_{p^k}$ for each prime $p$ and natural number $k$. The fact that the same L-function arises in these two different ways is a consequence of an important conjecture due to Serre [2,3]. This can as such be seen as a generalisation of the Taniyama-Weil conjecture, which, following on from the work of Wiles [4] and Taylor and Wiles [5], was proved by Breuil, Conrad, Diamond and Taylor [6]. Further work by Taylor, and many others, led to a complete proof of the Serre conjecture by Dieulefait [7], Khare and Wintenberger [8,9] and Kisin [10]. This proof is regarded as an important development in number theory.

The right hand side of this identity is a series expression, where the summand's most interesting inhabitants are the degree $k$ genus 0 Gromov–Witten invariants $N_k^{GW}$ of the compact one-parameter Calabi–Yau threefold defined by the intersection of two cubics in $\mathbf{P}^5$, which was argued in [11] to possess the necessary relation (modularity) to the L-function $L_{\mathbf{54.4.a.c}}(s)$. The subscript **54.4.a.c** gives the LMFDB label of the L-function [12]. The symbol $\mathfrak{p}$ denotes a partition of the integer $j$, and $l(\mathfrak{p})$ is the length of the partition $\mathfrak{p}$. The Gromov–Witten invariants enter in nonlinear combinations $\widetilde{N}_{\mathfrak{p}}$ which we define in (44), and $K_\nu$ is a modified Bessel function of the second kind. One can note that such Bessel functions of half odd integral order are the product of an exponential and a polynomial.

The principal aim of this paper is to provide more such identities, as we do in §4.2. They demonstrate an interesting and unexplored relationship between quantities defined separately in number theory and symplectic geometry. The identities provide a direct relation between the instanton numbers of a Calabi–Yau threefold defined over $\mathbf{C}$ and the collection of the point counts of that variety over each finite field $\mathbf{F}_{p^k}$.

These identities remain conjectural, although in the best case we are able to check them numerically to 130 figures. In the worst case, we can only verify the identity to six figures. We will explain in §5 why we believe that our worst examples are bad as a consequence of inadequacies of our resummation techniques, rather than simply being incorrect.

The first such identity was written down in [1]. At the time, more could not be given owing to a paucity of known rank-two attractors (which serve as necessary input for each such identity) and the lack of a way to make sense of the often divergent series that arise in this way.

In light of the works [11, 13], the situation is much improved and it is clear that many smooth nonrigid rank-two attractors, beyond the examples of [14], do in fact exist (at least conjecturally, on the strength of extensive computational data produced by the methods of [15]). We are able to add to this growing list, and in this paper document the existence of two new smooth attractor varieties.

One of our new attractors belongs to the mirror of the Reye-Congruence Calabi–Yau threefold, studied by Hosono and Takagi in the series of papers [16–19]. This model furnishes an important example in the GLSM literature, for instance see [20], as it is an example of a one-parameter Calabi–Yau threefold with two derived equivalent (but non-birational) mirror manifolds. One mirror can be obtained as the $\mathbf{Z}_2$ quotient of a certain complete intersection in $\mathbf{P}^4 \times \mathbf{P}^4$, and in fact our new attractor is also a rank-two attractor for the multiparameter manifold covering the mirror Reye-Congruence. That is to say, we also have a rank-two attractor in the two-parameter moduli space of the mirror of this complete intersection in $\mathbf{P}^4 \times \mathbf{P}^4$. Our second new attractor belongs to a less well-studied space on which we make some preliminary investigation.

The summation of divergent series is addressed by the use of Padé approximants, and we explain our attempts to extract as much numerical agreement between the left and right hand sides of our identities as we can manage, subject to the constraint that we are only able to compute 76 terms of the series owing to computational limitations. Our application of Padé approximants follows the textbook [21], and we find application for the conformal map techniques explained in [22].

After fixing some notation and introducing periods in §2.1, we give a brief review of some relevant number theory and geometry in §2.2. The identities originate in the attractor mechanism of 4d $\mathcal{N}=2$ supergravities obtained from Calabi–Yau compactification. Certain points in the moduli spaces of scalar fields, rank-two attractor points studied in [14, 23, 24], possess various interesting number-theoretic properties which provide the theoretical origin of our series. We review the attractor mechanism in §2.3. Our sums arise from applying the methods of [1] to solve the attractor equations in the special rank-two case, as reviewed in §2.4. §2.5 summarises the method that generates our identities. §2.6 fixes some notation that will appear when we consider weight-two modularity.

We go on to present our new attractors in §3, and there draw attention to the Beilinson–Bloch conjectures. We do so because one of the L-functions that arises has nonzero analytic rank. §4 and §5 respectively present our sums and describe the numerical methods we apply to our series.

# 2 Periods, modularity, and attractors

## 2.1 Periods

We study Calabi–Yau threefolds $X^\varphi$ with $h^{1,2}(X^\varphi) = 1$. Such threefolds have one complex structure modulus, which we denote by $\varphi \in \mathbf{P}^1 \setminus \{\Delta = 0\}$, where $\Delta = 0$ is the discriminant locus of the family.

A Picard–Fuchs equation governs the variation of complex structures for such a family $X^\varphi$. This is a fourth-order Fuchsian differential equation in the complex structure parameter $\varphi$, which in our examples will possess a point of maximal unipotent monodromy (MUM-point)

which can be taken to be at $\varphi = 0$. At this point the indicial equation of the operator has a root with multiplicity four. Frobenius bases of solutions, with a convenient scaling by powers of $2\pi i$ and a further normalisation, can be formed by taking series around $\varphi = 0$ with the following asymptotics:

$$
\begin{aligned}
\widehat{\varpi}_0 &= 1 + O(\varphi), \\
\widehat{\varpi}_1 &= \frac{\log(\varphi)}{2\pi i} + O\left(\varphi \log(\varphi)\right), \\
\widehat{\varpi}_2 &= \frac{Y_{111}}{2}\left(\frac{\log(\varphi)}{2\pi i}\right)^2 + O\left(\varphi \log(\varphi)^2\right), \\
\widehat{\varpi}_3 &= \frac{Y_{111}}{6}\left(\frac{\log(\varphi)}{2\pi i}\right)^3 + O\left(\varphi \log(\varphi)^3\right).
\end{aligned}
\tag{2}
$$

There will also be utility in what follows for sometimes removing the $2\pi i$ to recover the standard Frobenius basis

$$
\varpi = \nu \widehat{\varpi}, \qquad \text{with} \qquad \nu = \text{diag}(1, 2\pi i, (2\pi i)^2, (2\pi i)^3). \tag{3}
$$

Here $\varpi = (\varpi_j)$ and $\widehat{\varpi} = \left(\widehat{\varpi}_j\right)$ are column vectors. When the MUM point is mirror to the large volume point of a family $Y^t$ with Kähler parameter $t$, $Y_{111}$ is the triple intersection number of $Y^t$. With $e_1$ denoting the generator of $H^2(Y^t, \mathbf{Z})_{\text{Free}}$, $Y_{111}$ can be computed via

$$
Y_{111} = \int_{Y^t} e_1 \wedge e_1 \wedge e_1. \tag{4}
$$

We remark that in this work we have incorporated factors of $Y_{111}$ in the bases (2), (3), so that our presentation is uniform with the standard multiparameter expressions [25, 26], although sometimes in the literature such factors are left out of the Frobenius basis.

An additional important basis is motivated by the genus-0 prepotential for the A-model topological string theory, with target space a smooth Calabi–Yau manifold $Y^t$ with $h^{1,1}(Y^t) = 1$. This reads

$$
F^{(0)} = -\frac{1}{6}Y_{111}t^3 - \frac{1}{2}Y_{011}t^2 + \frac{c_2(Y)}{24}t + \frac{\zeta(3)}{(2\pi i)^3}\frac{\chi(Y)}{2} - \frac{1}{(2\pi i)^3}\sum_{k=1}^{\infty} n_k^{(0)} \text{Li}_3\left(e^{2\pi i k t}\right). \tag{5}
$$

The integer $c_2$ is computed from the second Chern class $c_2(Y)$ of the mirror $Y^t$ by

$$
c_2 = \int_{Y^t} c_2(Y^t) \wedge e_1, \tag{6}
$$

$Y_{011} = \frac{1}{2}(Y_{111} \bmod 2)$, and $\chi$ is the Euler characteristic of $Y^t$. The last term on the right hand side of (5) is the instanton sum, which encodes quantum corrections in the A-model. The $k^{th}$ term in the sum gives the contribution of degree-$k$ holomorphic rational maps from the string worldsheet into $Y^t$, so that the instanton numbers $n_k^{(0)}$ have an enumerative interpretation as the numbers[1] of such maps, up to subtleties discussed, for instance, in [28].

The Kähler parameter $t$ and complex structure parameter $\varphi$ can be related by the mirror map

$$
t(\varphi) = \frac{\widehat{\varpi}_1(\varphi)}{\widehat{\varpi}_0(\varphi)}. \tag{7}
$$

---

[1]Integrality of these numbers, including the higher genus $n_k^{(g)}$, was proved by symplectic geometry methods in [27].

From the prepotential (5), the integral symplectic period vector $\Pi(t)$ is formed via

$$\Pi(t) = \begin{pmatrix} 2F^{(0)} - t\partial_t F^{(0)} \\ \partial_t F^{(0)} \\ 1 \\ t \end{pmatrix}. \tag{8}$$

Using the mirror map (7), this prescribes another basis of periods. The change of basis from the Frobenius solutions (2) is effected by

$$\Pi = R\widehat{\varpi} = R\nu^{-1}\varpi, \qquad \text{with} \qquad R = \begin{pmatrix} \frac{\zeta(3)\chi}{(2\pi i)^3} & \frac{c_2}{24} & 0 & 1 \\ \frac{c_2}{24} & -Y_{011} & -1 & 0 \\ 1 & 0 & 0 & 0 \\ 0 & 1 & 0 & 0 \end{pmatrix}. \tag{9}$$

The solution basis $\Pi$ in (9) gives the integrals of the holomorphic three-form $\Omega$ over an integral symplectic basis of three-cycles, $\{A_0, A_1, B^0, B^1\}$ in $H_3(X^\varphi, \mathbf{Z})$ satisfying the symplectic relations

$$A_a \cap B^b = \delta_a^b, \qquad A_a \cap A_b = B^a \cap B^b = 0. \tag{10}$$

The dual cohomology basis $\{\alpha_0, \alpha_1, \beta^0, \beta^1\}$ of $H^3(X^\varphi, \mathbf{Z})$ satisfies

$$\int_{A_a} \alpha_b = \delta_b^a = -\int_{B^b} \beta^a, \tag{11}$$

with the other integrals vanishing. Such a basis is unique up to $\mathrm{Sp}(4, \mathbf{Z})$ transformations, which we can uniquely fix by requiring that through the mirror map the periods

$$z^a = \int_{A_a} \Omega, \qquad \mathcal{F}_b = \int_{B^b} \Omega, \tag{12}$$

map to the mirror periods (8). Special geometry relations [29] imply that there exists a B-model prepotential, a degree-2 homogenous function $\mathcal{F}(z)$ of half of the periods $z^a$, such that the remaining periods are computed by taking derivatives

$$\mathcal{F}_b = \frac{\partial \mathcal{F}}{\partial z^b}. \tag{13}$$

This basis of periods can be expressed via integrals as

$$\begin{pmatrix} \int_{B^0} \Omega \\ \int_{B^1} \Omega \\ \int_{A_0} \Omega \\ \int_{A_1} \Omega \end{pmatrix} = \begin{pmatrix} \mathcal{F}_0 \\ \mathcal{F}_1 \\ z^0 \\ z^1 \end{pmatrix}. \tag{14}$$

This above expression is equivalent to (9) through the identification $t = z^1/z^0$, use of the mirror map (7), a scaling transformation $\Pi \to \Pi/z^0$, and finally a coordinate choice $z^0 = 1$ (made only after taking the derivatives (13)). This manipulation, only briefly outlined here, has been explained many times before [26, 29] including in the present notation in [1].

Importantly, monodromies about singularities in moduli space effect symplectic transformations on the periods. For smooth Calabi–Yau target spaces, the period vector in the basis (12) undergoes *integral* symplectic transformations [29]. It has long been appreciated that this integrality property restricts the allowable set of topological data [26, 30], for instance

one always has $2Y_{011} = Y_{111} \mod 2$ as a consequence of the shift transformation $t \mapsto t + 1$ effecting an integral symplectic transformation on (8).

We will make use of the following choice of a rational basis,[2] which is related to the Frobenius periods through a simpler relation

$$\widetilde{\varpi}_i \;=\; \widehat{\varpi}_i + \delta_{i,3} \frac{\chi \zeta(3)}{(2\pi \mathrm{i})^3} \widehat{\varpi}_0 \,. \tag{15}$$

Use of this basis allows us to express periods in terms of L-values without introducing terms multiplied by $\zeta(3)/(2\pi \mathrm{i})^3$. This basis has the property that the monodromies of this basis of periods around the regular singular points of the corresponding Picard–Fuchs operators are rational.

## 2.2 Classical modularity of Calabi–Yau threefolds

Assuming that $X^{\varphi}$ is defined over $\mathbf{Q}$, meaning that it is defined as a zero locus of a number of polynomial equations with coefficients in $\mathbf{Q}$, then one can clear denominators so that the defining equations have integer coefficients. These equations can be reduced modulo a prime $p$ so that one can define the *local zeta function* $\zeta_p(X^{\varphi}, T)$ of the manifold $X^{\varphi}/\mathbf{Q}$ for each prime $p$:

$$\zeta_p(X^{\varphi}; T) \stackrel{\text{def}}{=} \exp\left( \sum_{n=1}^{\infty} N_{p^n}(X^{\varphi}) \frac{T^n}{n} \right), \tag{16}$$

where $N_{p^n}(X^{\varphi}) \stackrel{\text{def}}{=} \#X(\mathbf{F}_{p^n})$ is the number of points on $X^{\varphi}$ over $\mathbf{F}_{p^n}$. $\mathbf{F}_{p^n}$ denotes the finite field with $p^n$ elements, which has as a subfield $\mathbf{F}_p \cong \mathbf{Z}/p\mathbf{Z}$. Through this latter isomorphism the integer coefficients in the defining equations of $X^{\varphi}$ are embedded in $\mathbf{F}_{p^n}$.

From the Weil conjectures [31], proved in [32–34], it follows that $\zeta_p(X^{\varphi}, T)$ is in fact a rational function in the formal variable $T$. In particular, if the Calabi–Yau threefold $X^{\varphi}$ satisfies the technical assumption that its Picard group is generated by divisors defined over $\mathbf{F}_p$, the zeta function is given by the simple rational function

$$\zeta_p(X^{\varphi}; T) \;=\; \frac{R_p(X^{\varphi}; T)}{(1-T)(1-pT)^{h^{1,1}}(1-p^2 T)^{h^{1,1}}(1-p^3 T)} \,. \tag{17}$$

In one-parameter cases, the numerator $R_p(X^{\varphi}, T)$ is a polynomial, in $T$, of degree $b_3(X^{\varphi}) = 4$. For generic $\varphi$ valued in a number field these $R_p$, which do not necessarily factorise over $\mathbf{Q}$, are expected to encode the data of an automorphic form. In this way an automorphic L-function is associated to a motivic L-function, which is defined from the data of point counts $N_{p^n}(X^{\varphi})$. The motivic L-function is defined by an Euler product

$$L(X^{\varphi}; s) \;=\; \prod_{p \text{ bad}} B_p(X^{\varphi}; s) \prod_{p \text{ good}} \frac{1}{R_p(X^{\varphi}; p^{-s})} \,. \tag{18}$$

The finitely many factors $B_p(s)$ come from the primes of bad reduction, as explained in [35]. Identifying motivic and automorphic L-functions is a tenet of the Langlands correspondence, which has a relatively accessible introduction in [36] section 3.1, see also chapter one of [37].

Varieties $V$ defined as the set of points in $\mathbf{F}_{p^n}$ obeying polynomial constraints, where the polynomials have coefficients in $\mathbf{F}_p$, possess a certain automorphism: the Frobenius map

$$\text{Frob}_p : x \mapsto x^p \,. \tag{19}$$

---

[2]This is, up to a different choice of normalisation, the basis called *modified complex Frobenius basis* in [15].

This sends each coordinate $x$ of the ambient space to its $p^{\text{th}}$ power. To see that this provides an automorphism of $V$, suppose now that $F(x) = \sum c_{\mathbf{m}} x^{\mathbf{m}}$ is a multivariate polynomial in $x_{m_1} \dots x_{m_n}$ with integer coefficients $c_{\mathbf{m}}$ (where we use a multi index notation $x^{\mathbf{m}} = x_1^{m_1} \dots x_n^{m_n}$). In virtue of Fermat's Little Theorem, $c^p = c \bmod p$ for $c \in \mathbf{Z}$. Together with the fact that every non-unity multinomial coefficient in the expansion of $F(x)^p$ is a multiple of $p$, this has the consequence that for $x \in \mathbf{F}_{p^k}$

$$F(x)^p = F(x^p). \tag{20}$$

This relation also holds for $x$ in the algebraic closure $\mathbf{F}_p^{\text{alg}}$. From this it follows that

$$F(x) = 0 \iff F(x^p) = 0. \tag{21}$$

One sees from this that the set $F(x) = 0$ is preserved by the Frobenius map. Dwork [32] demonstrated that the zeta function of a (not necessarily Calabi–Yau) variety $X$ of complex dimension $d$ could be written as

$$\zeta_p(X;T) = \prod_{k=0}^{2d} \det\left(1 - T\,\text{Frob}_p^{(k)}\right)^{(-1)^{k+1}}. \tag{22}$$

By this formula, Dwork presented the zeta function as a rational function of $T$ as predicted by the Weil conjectures. $\text{Frob}_p^{(k)}$ is the induced action of the Frobenius map on the cohomology[3] $H^k$:

$$\text{Frob}_p^{(k)} : H^k \mapsto H^k. \tag{23}$$

For the case of smooth Calabi–Yau threefolds with $h^{2,1} = 1$ and $b_1 = b_5 = 0$, equation (22) reproduces (17). We wish to stress that the numerator $R_p(X^\varphi; T)$ in (17) is the determinant of the Frobenius action on the middle cohomology $H^3(X^\varphi)$.

Now it may happen that for some value $\varphi_*$ of the modulus, the action of the Frobenius map on the middle cohomology is block reducible. If this is so, the determinant will factorise. In fact, from the Hodge conjecture (by the argument briefly outlined in section 1.3 of [14] and subject to the assumptions made on the cycle $S$ therein) it follows that this occurs when the Hodge structure splits, meaning that we can write the middle cohomology $H^3(X^{\varphi_*}, \mathbf{Q})$ as a direct sum

$$H^3(X^{\varphi_*}, \mathbf{Q}) = \Lambda_{\text{attractor}} \oplus \Lambda_{\text{elliptic}}, \tag{24}$$

where

$$\Lambda_{\text{attractor}} \stackrel{\text{def}}{=} \left(H^{(3,0)}(X, \mathbf{C}) \oplus H^{(0,3)}(X, \mathbf{C})\right) \cap H^3(X, \mathbf{Z}), \tag{25}$$

$$\Lambda_{\text{elliptic}} \stackrel{\text{def}}{=} \left(H^{(1,2)}(X, \mathbf{C}) \oplus H^{(2,1)}(X, \mathbf{C})\right) \cap H^3(X, \mathbf{Z}). \tag{26}$$

With a block-reducible Frobenius action, the polynomial $R_p(X^{\varphi_*}; T)$ factorises over $\mathbf{Q}$ into two[4] quadratic factors (at least for all but finitely many bad primes):

$$\begin{aligned} R_p(X^{\varphi_*}; T) &= \left(1 - \alpha_p p T + p^3 T^2\right)\left(1 - \beta_p T + p^3 T^3\right) \\ &= \left(1 - \alpha_p(pT) + p(pT)^2\right)\left(1 - \beta_p T + p^3 T^3\right). \end{aligned} \tag{27}$$

By Serre's modularity conjecture [2,3], the coefficients $\alpha_p$ and $\beta_p$ appearing in this factorisation (27) are the Fourier coefficients of respectively weight-two and weight-four modular

---

[3]To be precise, this only works for any Weil cohomology theory.

[4]This factorisation need not occur over $\mathbf{Q}$, as explained in [38]. We proceed on the assumption that it does. This is caused by subtleties that arise in passing between the singular and étale cohomology theories. Our specific examples support this assumption.

forms for some congruence subgroups of SL(2, **Z**). Threefolds $X^{\varphi_*}$ for which these correspondences hold are said to be classically modular [39].

The subject of Calabi–Yau modularity has been nicely reviewed in [39, 40]. For a selection of careful treatments, discussing among other things methods for proving modularity, one has [41–45] and references therein.

The methods of [15] yield lists of $R_p$ that, for a handful of values of $\varphi_*$ across several families, support a conjecture that these $X^{\varphi_*}$ are indeed classically modular. These examples appear in [11, 13, 14].

These modular varieties have some interesting physical interpretations. They lead to simple formulae in terms of critical L-values for semiclassical black hole entropies and topological string free energies [14, 46], D-brane masses [1, 11], and certain Feynman integrals [47]. Beyond the one-parameter setting, interesting connections between weight-two modularity, supersymmetric flux vacua, and F-theory were conjectured in [38, 48], seeing extensive further evidence through detailed examples in [49]. The (modular) flux vacuum solutions were related to solutions of gauged $\mathcal{N} = 2$ supergravity theories in [50]. Further study of the connection between Hodge theory and supersymmetric flux vacua has been undertaken in [51]. Going beyond the case of threefolds, the modularity of fourfolds and related physical questions have been investigated in [52].

## 2.3 The attractor mechanism and modularity

Modular varieties are related to BPS black holes in $\mathcal{N} = 2$ supergravity theories obtained as compactifications of IIA/B supergravities on Calabi–Yau manifolds. In type IIA theory, the vector multiplet scalars encode the complexified Kähler structure moduli of the compactification manifold, whereas in type IIB they encode the complex structure moduli. Mirror symmetry relates the IIA compactification on a Calabi–Yau threefold $Y^t$ and the IIB compactification on its mirror $X^\varphi$.

The radial evolution of scalar fields in spherical black hole solutions of these theories can display attractor behaviour [53–55]: for static, supersymmetric black hole configurations, the radial evolution of the vector multiplet scalars according to the equations of motion forces them to take a value at the black hole horizon that is independent of sufficiently small perturbation at infinity.[5] The radial evolution gives an attractor flow in the complex structure moduli space of $X^\varphi$ (in the type IIB setting). The endpoint of this flow, called the attractor point depends only on the charge of the black hole, and the basin of attraction where the values of the scalar fields at infinity lie.

The black holes are charged under $U(1)^{b_3(X^\varphi)}$, with a choice of charges corresponding to an element $\gamma \in H^3(X^\varphi, \mathbf{Z})$ due to the charge quantisation conditions. Therefore, given a charge form $\gamma \in H^3(X^\varphi, \mathbf{Z})$, the attractor mechanism provides a corresponding point $\varphi$ in the complex structure moduli space $\mathcal{M}_{CS}$.

The attractor flow can be described as the gradient flow of the central charge of the supersymmetry algebra

$$\mathcal{Z}(\gamma) = e^{K/2} \int_{X^\varphi} \gamma \wedge \Omega, \tag{28}$$

which depends on $\varphi$. The moduli space Kähler potential $K$ is given via the relation

$$e^{-K} = i \int_{X^\varphi} \Omega \wedge \overline{\Omega}. \tag{29}$$

---

[5]For finite perturbations one can encounter a wall-crossing phenomenon whereby a different basin of attraction is entered.

In the gradient flow description, the attractor points are the local minima of $|\mathcal{Z}(\gamma)|$ in $\varphi$-space. The minima are divided into two classes according to whether $\mathcal{Z}(\gamma)$ is vanishing or not (for a physical interpretation of these different solutions, see for example [23, 24, 56–58]). There is some disagreement in the literature regarding the terminology for these different solutions, so in this paper, following [1], we call the condition for $\mathcal{Z}(\gamma)$ to have a local minimum with $\mathcal{Z}(\gamma) \neq 0$ the *alignment equations*, and the condition $\mathcal{Z}(\gamma) = 0$ the *orthogonality equations*. For us, following [23], an attractor point is a solution to the alignment equations.[6]

A rank-two attractor is a point $\varphi_* \in \mathcal{M}_{\mathbf{CS}}$ such that it is the attractor value for two linearly independent charge forms $\gamma_1$ and $\gamma_2$. We have denoted these rank-two attractors by $\varphi_*$, the same symbol used for classically modular varieties, because in fact rank-two attractors conjecturally give classically modular threefolds: the two three-forms $\gamma_1$ and $\gamma_2$ furnish a basis of

$$H^3(X^{\varphi_*}, \mathbf{Z}) \cap \left( H^{(3,0)}(X^{\varphi_*}, \mathbf{C}) \oplus H^{(0,3)}(X^{\varphi_*}, \mathbf{C}) \right). \tag{30}$$

Conjecturally, comparison isomorphisms relate this to a two-dimensional subspace of the middle étale cohomology. This latter subspace furnishes a two-dimensional Galois representation, so that $R_p(X^{\varphi_*}; T)$ factorises in the form (27) by Serre's modularity conjecture [2, 3]. Therefore, as was done in [14], we can search for rank-two attractors by computing the polynomials $R_p(X^{\varphi_*}; T)$ for the first few hundred primes. If $R_p(X^{\varphi_*}; T)$ factorises for all primes (possibly apart from a few bad primes), this strongly suggests that the point $\varphi_*$ is indeed a rank-two attractor point. More evidence for this can be obtained by numerically computing the period vector $\widetilde{\varpi}$ in the rational basis so that one can ascertain the integral charge vectors $Q_1, Q_2$ that give the components of $\gamma_1, \gamma_2$ in an integral cohomology basis.

## 2.4 Solving the attractor equations

In [54], the alignment equations were reformulated so that in the case of Calabi–Yau compactifications they take the following form: a Calabi–Yau threefold $X^{\varphi_*}$ corresponds to an attractor point $\varphi_* \in \mathcal{M}_{\mathbf{CS}}$ for a charge form $\gamma$ if there exists a complex constant $C$ such that

$$\gamma = \mathrm{Im}\left[ C\Omega(\varphi_*) \right]. \tag{31}$$

In the case of rank-two attractor points, instead of directly using the above equation, it turns out to be simpler to use the fact that, in the one-parameter case, if a $\varphi_*$ solves (31) for two independent $\gamma_1, \gamma_2$ then there will necessarily exist two further independent vectors $\gamma_3, \gamma_4$ such that [1]

$$\int_{X^{\varphi_*}} \gamma_3 \wedge \Omega = \int_{X^{\varphi_*}} \gamma_4 \wedge \Omega = 0. \tag{32}$$

Using the homology basis $\{A^0, A^1, B_0, B_1\}$, which satisfies (11), the components of the charge form $\gamma$ define the charge vector $Q$ as

$$Q = \begin{pmatrix} \int_{B^0} \gamma \\ \int_{B^1} \gamma \\ \int_{A_0} \gamma \\ \int_{A_1} \gamma \end{pmatrix}. \tag{33}$$

In terms of the integral basis of periods (8), the attractor equations (31) read

$$Q = \mathrm{Im}\left[ C\Pi(\varphi_*) \right], \tag{34}$$

---

[6]Note, however, that in the standard dynamical systems terminology the solutions with $\mathcal{Z}(\gamma) = 0$ are attractors.

for some complex constant $C$.

The IIB setup as just described gives a way of relating number-theoretic quantities to the attractor mechanism. We now turn attention to a IIA setup, and in so doing find a point of entry in our analysis for the genus-0 invariants of the mirror Calabi–Yau manifold.

In IIA compactifications on $Y^t$, there is a single vector multiplet whose scalar component is the Kähler modulus $t$. So we will again look at solutions of equation (31), but with a period vector

$$\Pi(t) = \begin{pmatrix} \frac{Y_{111}}{6}t^3 + \frac{c_2}{24}t + \frac{\chi(Y_t)\zeta(3)}{(2\pi i)^3} - 2\mathcal{I}(t) + t\mathcal{I}'(t) \\ -\frac{Y_{111}}{2}t^2 - Y_{110}t + \frac{c_2}{24} - \mathcal{I}'(t) \\ 1 \\ t \end{pmatrix}. \tag{35}$$

This form follows from (8). The instanton sum $\mathcal{I}(t)$ has an expansion

$$\mathcal{I}(t) = \frac{1}{(2\pi i)^3} \sum_{k=1}^{\infty} N_k^{GW} e^{2\pi i k \cdot t}, \tag{36}$$

and should be regarded as giving quantum corrections to the prepotential, which in turn gives quantum corrections to the Yukawa coupling. Here $N_k^{GW}$ is the degree-$k$ genus-0 Gromov–Witten invariant of $Y^t$.

Given a charge form $\gamma$, each of the equations (31), (32) determine some value of $t$. Were it not for the instanton sum $\mathcal{I}$, these equations would be a simple coupled set of algebraic equations for the real and imaginary parts of $t$, which can be solved to give the 'classical' solution $t_0 = x_0 + iy_0$. This simplicity is destroyed by the instanton series, so that those equations encode a complicated transcendental dependence of $t$ on the components of $Q$.

The solution of these equations by perturbation theory, for a restricted set of charge vectors $Q$, was undertaken in [1]. The charge vectors considered were of the form

$$Q_{D4} = \kappa \begin{pmatrix} \Lambda \\ \Upsilon \\ 0 \\ 1 \end{pmatrix}, \qquad Q_{D6} = \kappa \begin{pmatrix} \Lambda \\ \Upsilon \\ 1 \\ 0 \end{pmatrix}. \tag{37}$$

Here $\kappa$, $\kappa\Lambda$, and $\kappa\Upsilon$ can be any three integers (as required by charge quantisation). The subscripts are related to the interpretation of a charged black hole solution in supergravity as a bound state of even-dimensional D$p$-branes, with the most general integral charge vector having entries that give the numbers $q_D$ of those branes[7]

$$Q = \begin{pmatrix} q_{D0} \\ q_{D2} \\ q_{D6} \\ q_{D4} \end{pmatrix}. \tag{38}$$

Both of the vectors in (37) have integral components. The overall scale $\kappa \in \mathbf{Z}$ drops out of the equations we will solve, and we will solve for $t$ in terms of $\Lambda, \Upsilon \in \mathbf{Q}$. The solutions of [1] all take the form

$$t(\Lambda, \Upsilon) = t_0(\Lambda, \Upsilon) + \sum_{k=1}^{\infty} c_k\big(\Lambda, \Upsilon, y_0(\Lambda, \Upsilon)\big) e^{-2\pi k y_0(\Lambda, \Upsilon)}, \tag{39}$$

---

[7]One should bear in mind that we are displaying charge vectors of the supergravity theory. These can differ from the appropriate charge vectors of a dual microscopic theory owing to shifts induced by curvature couplings in D-brane worldvolume theories. This is explained in detail in [59], and reviewed in the present context in Appendix C of [1].

where $t_0(\Lambda, \Upsilon) = x_0(\Lambda, \Upsilon) + iy_0(\Lambda, \Upsilon)$ is the algebraic 'classical' solution to (31) or (32) when the instanton terms in (35) are ignored. The functional dependence of $t_0$ on the charges changes according to which equation is considered and which of the charge vectors (37) is used. The $c_k\big(\Lambda, \Upsilon, y_0(\Lambda, \Upsilon)\big)$ are rational functions of $y_0(\Lambda, \Upsilon)$ involving nonlinear combinations of the Gromov–Witten invariants. These invariants can be computed from the instanton numbers, using the methods of [29].

The particular solution $t(\Lambda, \Upsilon)$ that we use in this paper is that of

$$Q_{D4}^T \Sigma \Pi(t) = 0, \qquad \Sigma = \begin{pmatrix} 0 & \mathbf{I}_2 \\ -\mathbf{I}_2 & 0 \end{pmatrix}. \tag{40}$$

We choose to work with this equation because it belongs to a class for which the coefficients $c_k$ in the solution (39) are known to take a particularly simple form that uses Bessel functions [1]. Let us stress that we will be considering values of $t$ that solve the attractor equations (34) for two independent charge vectors $Q_1$ and $Q_2$, neither of which is the vector $Q_{D4}^T$ appearing in (40). One can see that $\Pi$ is a linear combination of $Q_1$ and $Q_2$. The subspace of $\mathbb{R}^4$ of vectors whose symplectic product with both of $Q_1, Q_2$ vanishes is two-dimensional, so there exists a basis $Q_3, Q_4$ of integer vectors so that $Q_3^T \Sigma \Pi = Q_4 \Sigma \Pi = 0$. Some suitable combination of $Q_3$ and $Q_4$ will be of the form $Q_{D4}$ in (37) with $q_{D6} = 0$.

This method does not provide new rank-two attractors, but organises the instanton contributions (Gromov–Witten invariants) in solutions of (40). By specialising to examples that we already know to be rank-two attractors (for instance based on the methods of [14]), we read off formulae that relate the Gromov–Witten invariants to number theoretical quantities.

Setting

$$x_0 = -\frac{Y_{110} + \Upsilon}{Y_{111}}, \qquad y_0 = \sqrt{\frac{2\Lambda - \frac{1}{12}c_2}{Y_{111}} - \left(\frac{Y_{110} + \Upsilon}{Y_{111}}\right)^2}, \tag{41}$$

the full solution to (40), given in [1], reads

$$t = x_0 + iy_0 - i\sum_{j=1}^{\infty} \frac{e^{2\pi ix_0 j}}{\sqrt{2\pi^3 Y_{111}}} \sum_{\mathfrak{p} \in \mathrm{pt}(j)} \widetilde{N}_{\mathfrak{p}} \left(\frac{j}{2\pi y_0 Y_{111}}\right)^{l(\mathfrak{p})-1/2} K_{l(\mathfrak{p})-1/2}(2\pi jy_0). \tag{42}$$

Here $\mathfrak{p}$ runs over partitions of an integer $j$, which we will write as

$$\mathfrak{p}: \qquad j = \sum_{k=1}^{j} \mu_k k. \tag{43}$$

For such a $\mathfrak{p}$ with multiplicities $\mu_k$ as above, we form the following products of Gromov–Witten invariants:

$$\widetilde{N}_{\mathfrak{p}} = \prod_{k=1}^{j} \frac{\left(kN_k^{GW}\right)^{\mu_k}}{\mu_k!}. \tag{44}$$

$l(\mathfrak{p})$ denotes the length of the partition $\mathfrak{p}$ and $K_\nu$ is the modified Bessel function of the second kind. The Bessel functions of half-integral order can be replaced with Bessel polynomials multiplied by exponentials, which is how one can express (42) in a form matching (39).

We study the equation (40) whose 'classical' part is quadratic and has therefore two solutions. One of these 'classical' solutions has a negative imaginary part, which implies that the perturbative expansion (39) that includes the instanton contributions necessarily diverges. Therefore, we discard this solution, leaving us with a unique 'classical' solution which leads to a convergent series for certain charges. With this in hand, we either sum the convergent series or study the resummation problem for the charge values in the case that the sum diverges.

## 2.5 Summary of our method

We obtain the series identities that give this paper its title as follows. First, we need a family $X^\varphi$ of Calabi–Yau threefolds with $h^{1,2} = 1$ that has a rank-two attractor point at some $\varphi_*$. Then there exist two linearly independent charge forms $\gamma_1$, $\gamma_2$ such that the equation (31) can be solved for $X^{\varphi_*}$. As discussed in the previous subsection, there also then exist two additional charge forms $\gamma_3$, $\gamma_4$ that satisfy the orthogonality equations (32). By taking a suitable linear combination of these, we can always find a solution for (40) for some pair $(\Lambda, \Upsilon)$. The explicit values for the charges $\Lambda$ and $\Upsilon$ are found numerically by analytically continuing the period vector $\Pi$ and evaluating it at the rank-two attractor point $\varphi_*$.

Each summation identity stems from a pair $(\varphi_{\mathrm{MUM}}, \varphi_*)$ of a MUM point and a rank-two attractor point for some family $X^\varphi$. This $\varphi_*$ will necessarily solve (40) for some pair $(\Lambda, \Upsilon)$ that we find. We assemble the enumerative invariants of the mirror $Y^t$ whose large volume point is mirror to $\varphi_{\mathrm{MUM}}$ into the series (42).

By conjectures due to Deligne [60], it is possible to express the periods $\widehat{\varpi}$ of a modular threefold in terms of special values of the associated L-function when those special functions are nonvanishing. By (7), the left hand side of (42) can therefore be expressed as a ratio of linear combinations of weight-four L-values. For a physicist-oriented account of Deligne's conjecture, one has [49, 61, 62].

## 2.6 Evaluating first derivatives of the periods

Following [14], we shall also provide relations between weight two critical L-values and covariant derivatives of the periods taken using the Kähler connection $K_\varphi$ as follows, using the Frobenius basis of periods displayed in (2):

$$D_\varphi \widehat{\varpi}_i \; = \; \partial_\varphi \widehat{\varpi}_i + K_\varphi \widehat{\varpi}_i, \quad \text{where}$$

$$K_\varphi \; = \; -\frac{\partial_\varphi \widehat{\varpi}^T \, \widehat{\sigma} \, \widehat{\varpi}^*}{\widehat{\varpi}^T \, \widehat{\sigma} \, \widehat{\varpi}^*}, \quad \text{with} \quad \widehat{\sigma} \; = \; \begin{pmatrix} \frac{2\chi\zeta(3)}{(2\pi i)^3} & 0 & 0 & -1 \\ 0 & 0 & 1 & 0 \\ 0 & -1 & 0 & 0 \\ 1 & 0 & 0 & 0 \end{pmatrix}. \tag{45}$$

It is important to note that the weight-two L-values make no appearance in the summation identities that give this paper its title.

Relations between Calabi–Yau periods and L-values have been systematically studied for the fourteen hypergeometric cases in [11], which presented two rank-two attractors in this set of geometries. Beyond solely studying the periods for the attractor varieties however, such relations were also observed (and even demonstrated rigorously in one case using a correspondence) for the conifold varieties. Moreover, the authors of [11] provide evaluations not just of the period vector but also all of its derivatives (which we do not do), using not only periods of modular forms (equivalent to the L-values that we use) but also quasiperiods of different meromorphic forms. These meromorphic forms are associated by the theory developed in [11] to the holomorphic forms prescribed by the $\mathbf{F}_{p^k}$ point-counts. In a similar vein, conifold modularity and its implications for period evaluations has been studied in [63] for families of double-octic threefolds. One should also see [64] for many results on periods, including identification of weight-three modular forms at K-points.

## 3 Two new rank-two attractors

The search method described in [14, 15] allows us to find rank-two attractor points in two distinct complex structure moduli spaces of Calabi–Yau manifolds. In the following, we refer to the Picard–Fuchs operators by their AESZ labels [65, 66].

We opt to present these rank-two attractors as special points in the complex structure moduli spaces of quotient manifolds with $h^{2,1} = 1$, mirror to different quotient manifolds with $h^{1,1} = 1$. However, these points also exist in the moduli space of the multiparameter manifolds with $h^{2,1} > 1$, which are simply connected covering spaces for the quotient geometries we present. This can be verified by briefly inspecting the attractor equations (34). One can therefore view our two new attractors (and indeed the attractors of [14]) as belonging to the moduli spaces of multiparameter manifolds, which may for some purposes be more useful.

### 3.1 AESZ22 / AESZ118

The operators AESZ22 and AESZ118 are related by a change of variables and a scaling as follows:

$$z = \frac{1}{32x}, \qquad \varpi_{22}(z) = \frac{1}{32x}\varpi_{118}(x). \tag{46}$$

AESZ22:

$$\begin{aligned}
\big[(1-32z)(7-4z)^2\big(1+11z-z^2\big)\theta_z^4 \\
-2z(7-4z)\big(143+4942z-2084z^2+256z^3\big)\theta_z^3 \\
-z\big(1638+102261z-72568z^2+23024z^3-3072z^4\big)\theta_z^2 \\
-z\big(637+66094z-30072z^2+12896z^3-2048z^4\big)\theta_z \\
-2z\big(49+7868z-1904z^2+1472z^3-256z^4\big)\big]\varpi_{22}(z) = 0,
\end{aligned}$$

AESZ118:

$$\begin{aligned}
\big[(1-x)(1-56x)^2\big(1-352x-1024x^2\big)\theta_x^4 \\
-2x(1-56x)(9-64x)\big(33+384x-1792x^2\big)\theta_x^3 \\
-x\big(431+15136x-335424x^2+4386816x^3-19267584x^4\big)\theta_x^2 \\
-2x\big(67+7072x-41088x^2+996532x^3-6422528x^4\big)\theta_x \\
-16x\big(1+176x-144x^2+23296x^3-20070x\big)\big]\varpi_{118}(x) = 0,
\end{aligned}$$

where $\theta_\varphi$ denotes the logarithmic derivative $\varphi\partial_\varphi$, and we have either $\varphi = z$ or $\varphi = x$ in each example.

AESZ22 has MUM points at $z = 0$ and $z = \infty$, which are respectively mapped to the MUM points $x = \infty$, $x = 0$ of AESZ118. Either of these operators can be taken as the Picard–Fuchs operator for a single family of Calabi–Yau threefolds[8] $X^z$. As was detailed in the papers [16, 17] by Hosono and Takagi, the point $z = 0$ is mirror to the large volume point of the CICY quotient

$$\mathbf{P}^4\begin{bmatrix} 1 & 1 & 1 & 1 & 1 \\ 1 & 1 & 1 & 1 & 1 \end{bmatrix}_{/\mathbf{Z}_2}, \tag{47}$$

where the freely acting $\mathbf{Z}_2$ symmetry exchanges the ambient $\mathbf{P}^4$ factors. This space is also known as the Reye Congruence, and this section is concerned with local zeta functions of the Mirror Reye Congruence as constructed in [16], which we denote in this section as $X^z$.[9]

---

[8]Here we have written $X^z$ with $z$ as the complex structure coordinate, but we could well have reparametrised and written $X^x$ with $z = \frac{1}{32x}$ which gives the same family.

[9]In [16] this was denoted by $X^\vee$.

An interesting result of [16, 17] is that the other MUM point $x = 0$ is mirror to the large volume point of a different family, given by smooth linear sections of the double quintic symmetroid, which is a double cover of the locus of $5 \times 5$ matrices of rank $\leq 4$ branched along the locus with rank $\leq 3$.

Our computations of the zeta function for members of this family $X^z$ give strong evidence, in the form of obvious persistent factorisations [14] of the zeta function, that

$$\boxed{z = -1 \quad \text{is a rank-two attractor.}} \tag{48}$$

The zeta function can be computed with either choice of Picard–Fuchs operator following the methods of [15], and is independent of this choice. The associated weight-two and weight-four modular forms are respectively

$$
\begin{aligned}
f_{\mathbf{11.2.a.a}}(\tau) &= q - 2q^2 - q^3 + 2q^4 + q^5 + 2q^6 - 2q^7 - 2q^9 + \dots, \\
f_{\mathbf{33.4.a.b}}(\tau) &= q - q^2 - 3q^3 - 7q^4 - 4q^5 + 3q^6 - 26q^7 + 15q^8 + 9q^9 + \dots,
\end{aligned}
\tag{49}
$$

where we have given the modular forms with their LMFDB labels[10] [12] and written $q = \mathrm{e}^{2\pi i \tau}$. Mellin transforming the above modular forms gives the L-functions $L_{\mathbf{33.4.a.b}}$ and $L_{\mathbf{11.2.a.a}}$, which have the critical values

$$
\begin{aligned}
L_{\mathbf{33.4.a.b}}(1) &= -1.0538249565444346000240168415\dots, \\
L_{\mathbf{33.4.a.b}}(2) &= \phantom{-}0, \\
L_{\mathbf{11.2.a.a}}(1) &= \phantom{-}0.25384186085591068433775892335\dots
\end{aligned}
\tag{50}
$$

The topological data of the Reye congruence

$$Y_{111} = 35, \qquad c_2 = 50, \qquad Y_{011} = \frac{1}{2}, \qquad \chi = -50, \tag{51}$$

can be used to fix the integral basis of periods $\Pi^{(0)}$ associated to the LCS point $z = 0$ of the operator AESZ22. Analogously, the topological data of the mirror at $z = \infty$

$$Y_{111} = 10, \qquad c_2 = 40, \qquad Y_{011} = 0, \qquad \chi = -50, \tag{52}$$

can be used to find an integral basis of periods $\Pi^{(\infty)}$ adapted to the point $z = \infty$.

These two integral bases are consistent in the sense that there exists an integral symplectic transfer matrix T such that [16]

$$\Pi^{(\infty)}(x) = \frac{1}{8x} \mathrm{T}\, \Pi^{(0)}(z(x)). \tag{53}$$

The matrix T is sensitive to the choice of path used to continue from small $z$, where series expressions for $\Pi^{(0)}(z)$ converge, to small $x$, where series expressions for $\Pi^{(\infty)}(x)$ converge. We use the contour displayed in Figure 1, beginning at small positive $z$, continuing into the upper half plane, moving straight to the left, and then down onto a large negative $z$. We use the principal branch of the logarithm to evaluate periods $\Pi^{(\infty)}(x)$ at the endpoint of this path. The matrix T is given by

$$
\mathrm{T} = \begin{pmatrix} 8 & 3 & -10 & 9 \\ -5 & -3 & 10 & -19 \\ 4 & 1 & -3 & -2 \\ 3 & 1 & -3 & 1 \end{pmatrix}. \tag{54}
$$

---

[10]It may be interesting to note that $f_{\mathbf{11.2.a.a}}$ is the lowest-level weight-two newform [12]. This weight-two form has previously appeared in [46] in the study of attractor points for AESZ101. This is expected to imply existence of a *correspondence* between the associated geometries, as we will discuss in some more detail in §3.2.

The factor of $\frac{1}{8x}$ in (53) amounts to a necessary Kähler transformation. The factor of $\frac{1}{x}$ is to be expected based on the scaling transformation used to transform AESZ22 to AESZ118. The extra factor of 1/8 is necessary to have T be symplectic, and ensures the correct large volume asymptotics for the Yukawa coupling.

In the rational basis $\widetilde{\varpi}^{(0)}$ adapted to the LCS point at $z = 0$, the periods at the attractor point $z = -1$ can be expressed in terms of L-values. However, note that $L_{\mathbf{33.4.a.b}}(2) = 0$ as per (50). The L-function has analytic rank 1, meaning that it has a first order zero at $s = 2$ (the central point in this case). As a consequence, there is no hope of expressing the (nonzero) real and imaginary parts of the Calabi–Yau periods in terms of L-values associated to the modular form $f_{\mathbf{33.4.a.b}}$ obtained by point counts. We will shortly turn to a discussion of theorems and conjectures that offer a way around this problem, and for now offer the following equalities that we have verified to 300 figures.

$$\widehat{\varpi}_0^{(0)}(-1) \overset{\text{def}}{=} \widetilde{\varpi}_0^{(0)}(-1) = -\frac{5}{3}\frac{L_{\mathbf{33.4.a.b}}(1)}{2\pi i} - \frac{5\sqrt{5}}{2}\frac{L_{\mathbf{825.4.a.f}}(2)}{(2\pi i)^2},$$

$$\widehat{\varpi}_1^{(0)}(-1) \overset{\text{def}}{=} \widetilde{\varpi}_1^{(0)}(-1) = -\frac{5\sqrt{5}}{4}\frac{L_{\mathbf{825.4.a.f}}(2)}{(2\pi i)^2},$$

$$\widehat{\varpi}_2^{(0)}(-1) \overset{\text{def}}{=} \widetilde{\varpi}_2^{(0)}(-1) = \frac{175}{36}\frac{L_{\mathbf{33.4.a.b}}(1)}{2\pi i} - \frac{25\sqrt{5}}{3}\frac{L_{\mathbf{825.4.a.f}}(2)}{(2\pi i)^2},$$

$$\widehat{\varpi}_3^{(0)}(-1) + \frac{\chi\,\zeta(3)}{(2\pi i)^3}\widehat{\varpi}_0^{(0)}(-1) \overset{\text{def}}{=} \widetilde{\varpi}_3^{(0)}(-1) = \frac{5}{6}\frac{L_{\mathbf{33.4.a.b}}(1)}{2\pi i} - \frac{25\sqrt{5}}{48}\frac{L_{\mathbf{825.4.a.f}}(2)}{(2\pi i)^2}.$$

(55)

Note that we have introduced a different L function, $L_{\mathbf{825.4.a.f}}$, which is obtained from $L_{\mathbf{33.4.a.b}}$ upon twisting by a Dirichlet character. Doing so has forced us to introduce the irrational algebraic number $\sqrt{5}$ in our period evaluations. The critical L-values of the new L-function are

$$L_{\mathbf{825.4.a.f}}(1) = 51.84133326282058811925374417867\ldots,$$
$$L_{\mathbf{825.4.a.f}}(2) = 2.98221153263635565613187 5081755\ldots$$

(56)

As will become clear once we state a theorem due to Shimura in the following subsection, we could also have opted to replace instances of $L_{\mathbf{33.4.a.b}}(1)$ in (55) with $L_{\mathbf{825.4.a.f}}(1)$, using

$$\frac{L_{\mathbf{33.4.a.b}}(1)}{2\pi i} = -\frac{\sqrt{5}}{110}\frac{L_{\mathbf{825.4.a.f}}(1)}{2\pi i}.$$

(57)

Due to multivaluedness, the exact expression for the periods depends on the choice of the contour used to analytically continue the periods from $z = 0$ to the rank-two attractor point at $z = -1$. We display the contour used to obtain the expression above in Figure 1.

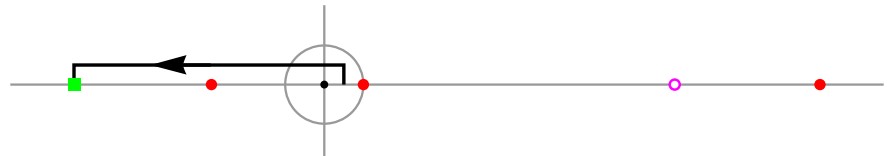

Figure 1: The complex $z$-plane for AESZ22. The blue square shows the rank-two attractor at $z = -1$. The three red discs display conifold points at $1/32$ and $\left(11 \pm 5\sqrt{2}\right)/2$. We use a magenta circle for the apparent singularity at $z = 7/4$. The black circle encloses the domain of convergence of the series expressions for $\widetilde{\varpi}^{(0)}$. This figure distorts distances, but we have preserved the ordering of the distances from each point of interest to the origin. The grey lines indicate the contour along which we continue the periods from $|z| < 1/32$ to $z = -1$.

**Shimura's theorem and the Beilinson–Bloch conjectures**

Following [12]: given a newform $f$ with Fourier expansion

$$f(\tau) = \sum_{n=1}^{\infty} a(n)q^n, \qquad q = e^{2\pi i \tau}, \tag{58}$$

and a primitive Dirichlet character $\phi$, one can construct a newform $f \otimes \phi$. This is the twist of $f$ by $\phi$, defined via the Fourier expansion

$$(f \otimes \phi)(\tau) = \sum_{n=1}^{\infty} b(n)q^n, \tag{59}$$

where $b(n) = \phi(n)a(n)$ for all $n$ coprime to both the level of $f$ and the conductor of $\phi$. This last relation only holds for the qualified $n$, but is sufficient to uniquely fix the newform $f \otimes \phi$.

Shimura's theorem, theorem 1 of [67],[11] states that given a cusp form $f \in S_k(N, \chi)$ of weight $k$ with character $\chi$, and a Dirichlet[12] character $\phi$, there exist two complex numbers $u^{\pm}$ such that

$$\frac{L_{f \otimes \phi}(m)}{(2\pi i)^m} \in \begin{cases} \mathfrak{g}(\phi)u^+ K_f K_\phi, & \text{if} \quad \phi(-1) = (-1)^m, \\ \mathfrak{g}(\phi)u^- K_f K_\phi, & \text{if} \quad \phi(-1) = (-1)^{m-1}, \end{cases} \tag{60}$$

for every positive integer $m < k$. Here $K_f$ denotes the number field generated over $\mathbf{Q}$ by the Fourier coefficients $a_n$ of $f$, $K_\phi$ denotes the field over $\mathbf{Q}$ generated by the numbers $\phi(n)$, $n \in \mathbf{Z}$. The function $L_{f \otimes \phi}$ is the $L$-function associated to the twist of $f$ by $\phi$. The Gauss sum $\mathfrak{g}(\phi)$ of the character $\phi$ is defined as the Gauss sum $\mathfrak{g}(\phi_0)$ of the associated primitive character $\phi_0$.

This theorem implies that if $f, g \in S_k(N, \chi)$ belong to the same twist orbit so that $g = f \otimes \psi$ for some primitive Dirichlet character $\psi$ then, if the twist $\psi$ has even parity (which means $\psi(-1) = 1$),

$$\frac{L_g(1)}{L_f(1)}, \frac{L_g(2)}{L_f(2)} \in \mathfrak{g}(\psi)K_f K_\psi. \tag{61}$$

If $\psi$ has odd parity (meaning that $\psi(-1) = -1$) then

$$(2\pi i)\frac{L_g(1)}{L_f(2)}, (2\pi i)^{-1}\frac{L_g(2)}{L_f(1)} \in \mathfrak{g}(\psi)K_f K_\psi. \tag{62}$$

---

[11]Which we learnt of from [68].

[12]Note that this character $\phi$ is not assumed to be primitive in this statement of Shimura's theorem.

In writing (61) and (62), we have assumed that the L-values in the denominators are nonzero. Importantly, because Gauss sums are algebraic, the left hand sides of (61) and (62) are algebraic numbers.

This theorem shows that it is not so surprising that a different L-function could be used in the period evaluations (55) than the one we obtained from point-counts. One could, if tempted, revisit [14] and replace every L-function with some twist, at the expense of introducing algebraic factors in the area-entropy evaluations therein. However, this is not a fully satisfactory explanation in our present case because $L_{\mathbf{33.4.a.b}}$ vanishes, and so the coefficient of proportionality in Shimura's theorem is trivially zero.

This problem, whereby an L-value can vanish, is circumvented in [11] through the use of periods of modular forms, which equal algebraic multiples of the nonzero L-values in the twist orbit.

We are very tempted to raise the subject of the Beilinson–Bloch conjectures, in the hope that AESZ22 and the mirror Reye Congruence may offer an interesting case study. The Beilinson–Bloch conjectures can be thought of as generalising Deligne's conjecture to cases where the critical value of the motivic $L$-function $L(M,s)$ vanishes. Here $M$ is a motive, and we briefly review this construction for the case where $M$ corresponds to $H^i(X, \mathbf{C})$ with $i$ odd and $X$ a smooth variety over $\mathbf{Q}$. The relevant case for us is $i = 3$ as in equation (18). We follow [69, 70], with [36, 71] providing nice background. Define $m = (i + 1)/2$, and denote by $CH^n(X)$ the Chow group of codimension $n$ cycles modulo rational equivalence and by $CH^n(X)_0$ the subgroup of cycles that are homologically trivial. The Beilinson–Bloch conjectures are as follows.

1. The order of vanishing of the motivic $L$-function at $s = m$ is given by the (conjecturally finite) dimension of $CH^n(X)_0$:

$$r_X \overset{\text{def}}{=} \mathrm{ord}_{s=m} L(M,s) = \dim_{\mathbf{Q}} CH^n(X)_0 \otimes \mathbf{Q}. \tag{63}$$

2. There exists a natural nondegenerate "height pairing"[13]

$$\langle *,* \rangle : CH^n(X)_0 \times CH^{\dim X+1-m}(X)_0 \to \mathbf{R}. \tag{64}$$

3. Denoting the determinant of this pairing by $R$, the leading coefficient of the Taylor expansion of $L(M,s)$ is related to Deligne's period $c_M(m)$ (and so to the periods of the Calabi–Yau manifold $X$) by

$$\lim_{s \to m} \frac{L(M,s)}{(s-m)^{r_X}} = \frac{L^{(r_X)}(M,s)}{r_X!} = Q\, c_M(m) R, \qquad \text{with} \qquad Q \in \mathbf{Q}. \tag{65}$$

These conjectures generalise the Birch–Swinnerton-Dyer conjectures [72] to varieties beyond elliptic curves. Birch and Swinnerton-Dyer conjectured that the order of vanishing at $s = 1$ of the L-function attached to an elliptic curve equals the rank of that curve, which the above item 1 generalises. Further, their famous conjectural formula for the first nonzero derivative at $s = 1$ is generalised by item 3. The third part of the conjecture (65) suggests that we could express our Calabi–Yau periods (55) using the nonvanishing $L'_{\mathbf{33.4.a.b}}(2)$, if we divided through by the regulator $R$ that comes from the height pairing (64). Unfortunately it is beyond us to compute this regulator at this time.

Our evaluations (55), together with the Beilinson–Bloch conjectures, imply some relation to complement (57) of the form

$$\frac{L'_{\mathbf{33.4.a.b}}(2)}{(2\pi \mathrm{i})^2 R} = l \sqrt{5}\, \frac{L_{\mathbf{825.4.a.f}}(2)}{(2\pi \mathrm{i})^2}, \qquad \text{for some } l \in \mathbf{Q}. \tag{66}$$

---

[13]As explained in [69], there are several proposed definitions for this pairing.

Similar relations between derivatives of L-values and L-values in the weight two case can be obtained by assuming the Birch–Swinnerton-Dyer conjecture for elliptic curves, and carefully studying how the BSD quantities transform upon a twist of the elliptic curve (in the sense of twisting varieties). For the weight-two rank-one case, rigorous proofs of such equalities follow from results due to Gross-Zagier [73] (Theorem 6.3 therein).

**The weight-two piece**

For completeness, we display here briefly the data related to the derivatives of the periods.[14] These are related to weight-two $L$-functions and will not feature in our summation identities.

The Kähler connection at the rank-two attractor point is given by

$$K_z^{(0)}(-1) = -\frac{2}{3}. \tag{67}$$

For the covariant derivatives of the periods, we find

$$D_z\widehat{\varpi}_0^{(0)}(-1) = \frac{L_{\mathbf{11.2.a.a}}(1)}{(2\pi i)^2}\left(\frac{25}{3} - \frac{125}{6}\frac{i}{u^\perp}\right),$$

$$D_z\widehat{\varpi}_1^{(0)}(-1) = \frac{L_{\mathbf{11.2.a.a}}(1)}{(2\pi i)^2}\left(\frac{25}{6} - \frac{25}{4}\frac{i}{u^\perp}\right),$$

$$D_z\widehat{\varpi}_2^{(0)}(-1) = \frac{L_{\mathbf{11.2.a.a}}(1)}{(2\pi i)^2}\left(\frac{725}{18} - \frac{875}{18}\frac{i}{u^\perp}\right), \tag{68}$$

$$D_z\widehat{\varpi}_3^{(0)}(-1) + \frac{\chi\,\zeta(3)}{(2\pi i)^3}D_z\widehat{\varpi}_0^{(0)}(-1) = \frac{L_{\mathbf{11.2.a.a}}(1)}{(2\pi i)^2}\left(\frac{575}{72} - \frac{125}{16}\frac{i}{u^\perp}\right).$$

Here $u^\perp$ is a real constant with decimal expansion

$$u^\perp = 1.087533286862971250700\ldots \tag{69}$$

This is related to the elliptic curve with LMFDB label **11.a3** (see [14,49] for an explanation of this). The reduced minimal Weierstrass model is

$$y^2 + y = x^3 - x^2, \tag{70}$$

and the $j$-invariant of this elliptic curve is

$$j\left(\frac{1}{2} + iu^\perp\right) = -\frac{4096}{11}. \tag{71}$$

Note that due to the freedom of changing the rational basis of periods, the constant $u^\perp$ is only defined up to an overall rational constant. However, as discussed in [49], the curves associated to different values of $u^\perp$ related by such a scaling are isogenous and have in particular the same zeta functions. By going to the integral basis and fixing $u^\perp$ as a lattice parameter of the lattice generated over $\mathbf{Z}$ by $D_z\Pi$ and $\overline{D_z\Pi}$, we get a canonical choice for the parameter.

---

[14]These are related to motives of Hodge type $(1,0)+(0,1)$ by a Tate twist. These Tate twisted motives can be realised as motives associated to the middle cohomology of an elliptic curve as exemplified by the relations in this section.

## 3.2 AESZ17 / AESZ290

We also study another such pair of operators related by the transformations

$$z = -\frac{1}{729x}, \qquad F_{17}(z) = -\frac{1}{729x}F_{290}(x), \tag{72}$$

AESZ17:

$$\begin{aligned}
\Big[(1-27z)(5-9z)^2\left(1+27z^2\right)\theta_z^4 & \\
-36z\left(5-9z\right)\left(7-15z+621z^2-729z^3\right)\theta_z^3 & \\
-6z\left(180-541z+39591z^2-91935z^3+59049z^4\right)\theta_z^2 & \\
-6z\left(75-155z+34155z^2-64233z^3+39366z^4\right)\theta_z & \\
-3z\left(25-30z+21060z^2-32562z^3+19683z^4\right)\Big]F_{17}(z) &= 0,
\end{aligned}$$

AESZ290:

$$\begin{aligned}
\Big[(1+27x)(1+405x)^2(1+19683x^2)\theta_x^4 & \\
-108x\left(1+405x\right)\left(7-729x-177147x^2-7971615x^3\right)\theta_x^3 & \\
-6x\left(80-37017x-8155323x^2-1506635235x^3-87169610025x^4\right)\theta_x^2 & \\
-6x\left(17-17415x-3720087x^2-789189885x^3-58113073350x^4\right)\theta_x & \\
-9x\left(1-1998x-454896x^2-111602610x^3-9685512225x^4\right)\Big]F_{290}(x) &= 0.
\end{aligned}$$

Both $z=0$ and $x=0$ are points of maximal unipotent monodromy. Monodromy computations for AESZ17 were undertaken in [74]. The large complex structure point at $z=0$ is mirror to the large volume point of

$$\begin{array}{c}
\mathbf{P}^2 \\ \mathbf{P}^2 \\ \mathbf{P}^2
\end{array}
\left[\begin{array}{ccc}
1 & 1 & 1 \\
1 & 1 & 1 \\
1 & 1 & 1
\end{array}\right]_{/\mathbf{Z}_3}. \tag{73}$$

Before taking the $\mathbf{Z}_3$ quotient, the above complete intersection is an intersection of three hypersurfaces in the toric variety $\mathbf{P}^2 \times \mathbf{P}^2 \times \mathbf{P}^2$. The mirror variety can then be determined via a combinatoric procedure. We follow [26], which in turn makes reference to the original procedures of [75–77]. This leads to a complete intersection variety[15] with three complex structure parameters:

$$\begin{aligned}
1 - U_1 - V_1 - W_1 &= 0, \\
1 - U_2 - V_2 - W_2 &= 0, \\
1 - \frac{\varphi_1}{U_1 U_2} - \frac{\varphi_2}{V_1 V_2} - \frac{\varphi_3}{W_1 W_2} &= 0.
\end{aligned} \tag{74}$$

Upon setting $\varphi_1 = \varphi_2 = \varphi_3 = z$, we can identify a $\mathbf{Z}_3$ symmetry generated by

$$\begin{aligned}
U_1 &\mapsto V_1 \mapsto W_1 \mapsto U_1, \\
U_2 &\mapsto V_2 \mapsto W_2 \mapsto U_2,
\end{aligned} \tag{75}$$

which is fixed-point free for $z \neq \frac{1}{27}$ (which is on the singular locus). Taking the $\mathbf{Z}_3$ quotient gives us a manifold[16] $X^z$, mirror to the quotient (73), whose arithmetic properties are the topic of this section.

---

[15]Here the $U_i$, $V_i$, $W_i$ are coordinates on the algebraic torus $(\mathbf{C}^*)^6$ that is dense in the toric variety defined by the Batyrev-Borisov procedure, starting from the polyhedron data of (73) before the $\mathbf{Z}_3$ quotient is taken.

[16]This section is independent of the previous one, and so we hope that no confusion arises by our recycling of the notation $X^z$.

Computing the zeta function for $X^z$, we find evidence that

$$\boxed{z = -1 \quad \text{is a rank-two attractor.}} \tag{76}$$

We do not attach any significance to the fact that this is the same attractor value as found for the pair AESZ22/118. This time the associated modular forms, whose coefficients appear in (27), are

$$
\begin{aligned}
f_{\mathbf{14.2.a.a}}(\tau) &= q - q^2 - 2q^3 + q^4 + 2q^6 + q^7 - q^8 + q^9 + \dots, \\
f_{\mathbf{14.4.a.b}}(\tau) &= q + 2q^2 - 2q^3 + 4q^4 - 12q^5 - 4q^6 + 7q^7 + 8q^8 - 23q^9 + \dots
\end{aligned}
\tag{77}
$$

We note in passing the fact that the above weight-two form $f_{\mathbf{14.2.a.a}}$ is the same one that arises for the rational rank-two attractor of AESZ34 studied in [14]. Moreover, both of the above modular forms arise at the rank-two attractor point of AESZ100 studied in [46], where the coincidence of the weight-two forms for AESZ34 and AESZ100 is also noted. The matching of these modular forms might be explained by the existence of isomorphic two-dimensional Galois representations for the middle cohomology of each variety. For each pair of modular varieties possessing these isomorphic Galois representations, by the Tate conjecture (see for instance [78,79]) there should exist a correspondence (an algebraic cycle in the product of the pair of varieties, see e.g. [80]) that induces the isomorphisms of the Galois representations.[17]

The critical $L$-function values are

$$
\begin{aligned}
L_{\mathbf{14.4.a.b}}(1) &= 0.8147623501326139670662512393 26\dots, \\
L_{\mathbf{14.4.a.b}}(2) &= 1.1362033857191110956218584623126\dots, \\
L_{\mathbf{14.2.a.a}}(1) &= 0.3302236593444805390282619 46122\dots
\end{aligned}
\tag{78}
$$

We again emphasise that the weight-2 $L$-function $L_{\mathbf{14.2.a.a}}$ will not feature in our construction of summation identities, but is included here as part of our discussion of the variety $X^{z=-1}$.

The topological data of the mirror manifold (73) is given by

$$
Y_{111} = 30, \qquad c_2 = 36, \qquad Y_{011} = 0, \qquad \chi = -30. \tag{79}
$$

Using the Euler characteristic above we can find the rational basis of periods $\widetilde{\varpi}$ in terms of which, based on a numerical computation to 300 figures, we can express the periods as

$$
\begin{aligned}
\widehat{\varpi}_0^{(0)}(-1) &\overset{\text{def}}{=} \widetilde{\varpi}_0^{(0)}(-1) = -14 \frac{L_{\mathbf{14.4.a.b}}(2)}{(2\pi i)^2}, \\
\widehat{\varpi}_1^{(0)}(-1) &\overset{\text{def}}{=} \widetilde{\varpi}_1^{(0)}(-1) = -7 \frac{L_{\mathbf{14.4.a.b}}(2)}{(2\pi i)^2} - \frac{3}{4} \frac{L_{\mathbf{14.4.a.b}}(1)}{2\pi i}, \\
\widehat{\varpi}_2^{(0)}(-1) &\overset{\text{def}}{=} \widetilde{\varpi}_2^{(0)}(-1) = -42 \frac{L_{\mathbf{14.4.a.b}}(2)}{(2\pi i)^2} - \frac{45}{4} \frac{L_{\mathbf{14.4.a.b}}(1)}{2\pi i}, \\
\widehat{\varpi}_3^{(0)}(-1) + \frac{\chi\,\zeta(3)}{(2\pi i)^3}\widehat{\varpi}_0^{(0)}(-1) &\overset{\text{def}}{=} \widetilde{\varpi}_3^{(0)}(-1) = -\frac{7}{2} \frac{L_{\mathbf{14.4.a.b}}(2)}{(2\pi i)^2} - \frac{19}{8} \frac{L_{\mathbf{14.4.a.b}}(1)}{2\pi i},
\end{aligned}
\tag{80}
$$

with the contour of integration used to obtain these displayed in Figure 2

---

[17]We thank Pyry Kuusela for discussion on this matter.

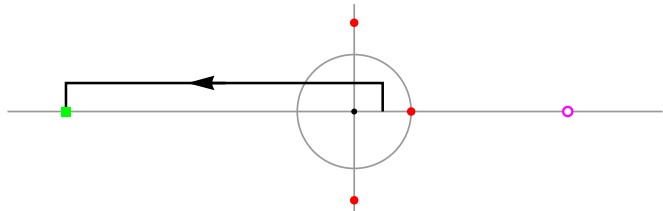

Figure 2: The complex $z$-plane for AESZ17. The green square shows our attractor $z = -1$. The three red discs display conifold points at $1/27$ and $\pm\sqrt{-3}/9$. We use a magenta circle for the apparent singularity at $z = 5/9$. The gray circle encloses the domain of convergence of the series expressions for $\widetilde{\varpi}^{(0)}$. This figure distorts distances, but we have preserved the ordering of the distances from each point of interest to the origin. The black lines indicate the contour along which we continue the periods from $|z| < 1/27$ to $z = -1$.

However, when we try to express these periods in the basis adapted to the rational basis associated to the MUM point at $z = \infty$, we run into trouble: using the Euler characteristic $\chi = -30$ and computing the period vectors the rational basis $\widetilde{\varpi}^{(\infty)}$, we find that the matrix T relating the bases $\widetilde{\varpi}^{(0)}$ and $\widetilde{\varpi}^{(\infty)}$ via

$$\widetilde{\varpi}^{(\infty)}(x) = \frac{1}{x} \mathrm{T}\, \widetilde{\varpi}^{(0)}\big(z(x)\big), \tag{81}$$

is not rational or symplectic, but is given by

$$\mathrm{T} = \sqrt{-3} \begin{pmatrix} \frac{-1}{54} & \frac{8}{81} & \frac{-1}{81} & \frac{2}{81} \\ \frac{-7}{972} & \frac{5}{162} & \frac{-1}{486} & 0 \\ \frac{25}{8424} & 0 & \frac{-5}{4212} & \frac{5}{2106} \\ \frac{25}{16848} & \frac{-25}{8424} & \frac{5}{25272} & 0 \end{pmatrix}.$$

$$\text{Note} \quad \mathrm{T}^T \begin{pmatrix} 0 & 0 & 0 & -1 \\ 0 & 0 & 1 & 0 \\ 0 & -1 & 0 & 0 \\ 1 & 0 & 0 & 0 \end{pmatrix} \mathrm{T} = \frac{5}{13 \cdot 3^8} \begin{pmatrix} 0 & 0 & 0 & -1 \\ 0 & 0 & 1 & 0 \\ 0 & -1 & 0 & 0 \\ 1 & 0 & 0 & 0 \end{pmatrix}. \tag{82}$$

The exact form the matrix T takes depends of course on the choice of path that we use to pass from small $z$ to small $x$. To arrive at the matrix above, we started at small positive $z$, move up into the upper half plane, move left underneath the conifold $x = \frac{\mathrm{i}}{3\sqrt{3}}$, and then down to a large negative $z$ (small positive $x$) as displayed in Figure 2.

This matrix allows us to express the rational basis of periods at infinity in terms of L-values, and we obtain

$$\widetilde{\varpi}_0^{(\infty)}\left(\tfrac{1}{729}\right) = \frac{9}{2}\,\mathrm{i}\sqrt{3}\frac{L_{14.4.a.b}(1)}{2\pi\mathrm{i}},$$

$$\widetilde{\varpi}_1^{(\infty)}\left(\tfrac{1}{729}\right) = -21\,\mathrm{i}\sqrt{3}\frac{L_{14.4.a.b}(2)}{(2\pi\mathrm{i})^2},$$

$$\widetilde{\varpi}_2^{(\infty)}\left(\tfrac{1}{729}\right) = \frac{45}{8}\,\mathrm{i}\sqrt{3}\frac{L_{14.4.a.b}(1)}{2\pi\mathrm{i}}, \tag{83}$$

$$\widetilde{\varpi}_3^{(\infty)}\left(\tfrac{1}{729}\right) = -\frac{315}{52}\,\mathrm{i}\sqrt{3}\frac{L_{14.4.a.b}(2)}{(2\pi\mathrm{i})^2}.$$

This is a surprising result, as the $\mathrm{i}\sqrt{3}$ cannot be explained away by an application of Shimura's theorem. $f_{\mathbf{14.4.a.b}}$ is the modular form obtained from point-counts (and not any

twist thereof), and indeed there is no $i\sqrt{3}$ in (80). In every example to date, periods in this rational basis evaluated at a rank-two attractor have been **Q**-multiples of the L-values associated to the relevant modular form.

We note an additional problem with this MUM point: we are not yet able to compute a meaningful value of the triple intersection number $Y_{111}$. Usually, by assuming an A-model prepotential of the form (5) one expects the basis provided by (12) to have integral monodromies. We are only able to obtain rational monodromies about conifolds, so that no entry of the monodromy matrix is a **Q**-multiple of $\zeta(3)/(2\pi i)^3$, if we take

$$Y_{111} = -\frac{30}{13}. \tag{84}$$

Clearly this is meaningless as a triple intersection number. Nonetheless we obtain a series identity in §4 by adopting this number. Moreover, the method of [15] only works in this case, using the Frobenius basis attached to the MUM point $x = 0$, if we work with $-30/13$ in place of the usual triple intersection number. We remark that it has recently been understood in [81] that certain MUM points require a modification to the A-model prepotential, which could explain our discrepancy. Naively proceeding with $Y_{111} = -30/13$ is sufficient for our present purposes of finding a (convergent) series identity for the relevant L-values.

**The weight-two piece**

We again study the derivatives of the periods in this subsection for completeness. However, we will not make further use of these identities in the present paper.

We find the following values for the Kähler connections:

$$K_z^{(17)}(-1) = -\frac{3}{4}, \qquad K_x^{(290)}\left(\tfrac{1}{729}\right) = \frac{729}{4}. \tag{85}$$

For the covariant derivatives of the periods, we find

$$D_z\widehat{\varpi}_0^{(17)}(-1) = \frac{L_{\mathbf{14.2.a.a}}(1)}{(2\pi i)^2}\left(\frac{27}{2}\right),$$

$$D_z\widehat{\varpi}_1^{(17)}(-1) = \frac{L_{\mathbf{14.2.a.a}}(1)}{(2\pi i)^2}\left(\frac{27}{4} + \frac{9}{8}\frac{i}{v^\perp}\right),$$

$$D_z\widehat{\varpi}_2^{(17)}(-1) = \frac{L_{\mathbf{14.2.a.a}}(1)}{(2\pi i)^2}\left(\frac{117}{2} + \frac{135}{8}\frac{i}{v^\perp}\right), \tag{86}$$

$$D_z\widehat{\varpi}_3^{(17)}(-1) + \frac{\chi\,\zeta(3)}{(2\pi i)^3}D_z\widehat{\varpi}_0^{(17)}(-1) = \frac{L_{\mathbf{14.2.a.a}}(1)}{(2\pi i)^2}\left(\frac{99}{8} + \frac{81}{16}\frac{i}{v^\perp}\right).$$

In this case the number $v^\perp$ has decimal expansion

$$v^\perp = 1.121098670864189300993\ldots, \tag{87}$$

and is related to the elliptic curve with LMFDB label **14.a5**. The reduced minimal Weierstrass equation is

$$y^2 + xy + y = x^3 - x. \tag{88}$$

The $j$-invariant of this curve is

$$j\left(\frac{1}{2} + iv^\perp\right) = -\frac{15625}{28}. \tag{89}$$

As in §3.1 the parameter $v^\perp$ is defined only up to a multiplication by a rational constant, and the associated elliptic curve is only defined up to an isogeny. Of course, if we multiply $v^\perp$ by a rational number, the right-hand side above changes accordingly as discussed in detail in [49].

# 4 L-value ratios from the master formula

In this section, we discuss specialising the 'master formula' (42) to a number of rank-two attractors. From the mirror map (7), together with the predictions of Deligne's conjecture [60], it follows in these cases that the $t$ on the left hand side is expressible in terms of the ratio $2\pi i L(1)/L(2)$ of critical values of the weight-four $L$-function associated to the attractor variety. In this way, we obtain interesting relations between the Gromov–Witten invariants and the L-function values. We do this for the attractors discussed in §3, and those appearing in [11,14]. In this way we obtain identities that are of the form

$$2\pi i \frac{L(1)}{L(2)} = \sum_{j=0}^{\infty} \sum_{\mathfrak{p}\in\mathrm{pt}(j)} \widetilde{N}_{\mathfrak{p}} A_{\mathfrak{p}}(\Lambda, \Upsilon), \tag{90}$$

where $\widetilde{N}_{\mathfrak{p}}$ are the products of Gromov–Witten invariants defined in (44), and $A_{\mathfrak{p}}(\Lambda, \Gamma)$ are simple functions related to modified Bessel functions of the second kind in a trivial way. We display the full identities for each rank-two attractor point in §4.2.

While such identities have already been discussed in [1], we are able to provide a larger set of examples by considering charge ratios $\Lambda_*, \Upsilon_*$ for which the series (42) does not converge. We are able to remedy this by analytically continuing the series (42) by using Padé resummation.

We wish to highlight a difference of the identities of the form (90) from the identities presented in [14] that express the value of the prepotential in terms of the L-function values, or higher genus prepotentials in terms of modular form periods and quasiperiods as in [46]. The main difference is that while those formulae are close to the form (90), with the functions $A_{\mathfrak{p}}$ being simple exponentials, in these cases the $A_{\mathfrak{p}}$ depend on the value of the flat coordinate $t$ at the rank-two attractor point. Since this value already is expressed in terms of special values of the $L$-function, we have that in such identities $L$-function values appear in a non-trivial way on both left- and right-hand sides of the resulting identity. This is to be contrasted with the formulae of the type (90) we study where the right-hand side summands only contain simple algebraic numbers and Bessel functions thereof.

## 4.1 Additional solutions via analytic continuation

The case where the series appearing in (90) converge was studied in [1]. In many cases however, the charge ratios $(\Lambda_*, \Upsilon_*)$ corresponding to a rank-two attractor point are such that the right-hand-side (90) is a divergent series. We shall discuss (although not rigorously prove) the existence of an analytic continuation, so that we can solve the orthogonality equation for any charge values. In order to evaluate this analytic continuation we appeal to Padé approximation, and so we recall some relevant theory of Padé approximants following the textbook [21].

We make a change of variables

$$(\Lambda, \Upsilon) \mapsto (x_0, y_0), \tag{91}$$

where

$$x_0 = -\frac{Y_{011}+\Upsilon}{Y_{111}}, \qquad y_0 = \sqrt{\frac{2\Lambda-\frac{1}{12}c_2}{Y_{111}}-\left(\frac{Y_{110}+\Upsilon}{Y_{111}}\right)^2}. \tag{92}$$

Let us now fix some value for $y_0$ and consider the function $G : \mathbf{C}^2 \mapsto \mathbf{C}$ defined by

$$G(x_0, t) \stackrel{\text{def}}{=} Q_{D4}^T \Sigma \Pi(t), \qquad \text{where} \qquad Q_{D4}^T \stackrel{\text{def}}{=} (\Lambda(x_0, y_0), \Upsilon(x_0, y_0), 0, 1). \tag{93}$$

Note that $G(x_0, t)$ is a quadratic function of $x_0$. Using square roots one can solve the orthogonality equation

$$G(x_0, t) = 0, \tag{94}$$

for $x_0$, which gives a function $x_0(t)$. Explicitly this is

$$x_0(t) = t + \sqrt{-y_0^2 - \frac{2}{Y_{111}} \mathcal{I}'(t)}. \tag{95}$$

We have chosen the positive branch of the square root above. With a fixed value for $y_0$ (so a fixed relationship between the charge ratios $\Lambda$ and $\Upsilon$), one can consider the problem of solving for the $t(x_0(\Upsilon))$ defined implicitly by (95). As a problem in complex analysis, the existence and properties of such solutions are sensitive to the singularity structure of $\mathcal{I}(t)$. We make working assumptions that an inverse function $t(x_0)$ exists for the charges of interest, and admits analytic continuation to our selected charges from within the radius of convergence of the series solution (42).

We do not provide anything like a closed form expression for the function $t(x_0)$, but proceed to approximate its analytic continuation with Padé approximants. These Padé approximants can be constructed from the data of the series coefficients in (42). Setting this up involves a further change of variables

$$\xi = \exp(2\pi i x_0). \tag{96}$$

Given a fixed value of $y_0$, defined by the charge ratios $\Lambda, \Upsilon$ from (40), the series solution (42) provides

$$t(\xi) = \sum_{j=0}^{\infty} c_j \xi^j. \tag{97}$$

A careful study of the coefficients $c_j$ demonstrates that this power series (97) has a finite radius of convergence in the $\xi$-plane [1]. While there certainly exist functions defined by convergent series that do not admit analytic continuation beyond their radii of convergence (for instance, the $q$-series of theta functions), we assume this is not the case for our functions. This assumption will be justified in-post by the successful evaluation of our Padé approximants, which are observed to better approximate the expected value with increasing order.

We then form Padé approximants to the series (97) in the variable $\xi$ of increasing order. We choose to use diagonal Padé approximants,[18] so that our order $d$ approximant is a ratio of degree $d$ polynomials in $\xi$ such that this rational function's Taylor series's first $2d$ terms agree with the series (97).

Note that, from our assumptions, our function $t(\xi)$ admits a single-valued analytic continuation to the whole of the complex plane, minus any singularities and branch cuts. There is some freedom in how to position these branch cuts while still having a single-valued continuation, but a specific positioning is singled out by Stahl's extremal domain theorem[19] [82]. Away from the cuts, convergence of the sequence of diagonal Padé approximants to the function's value is guaranteed by Stahl's Padé convergence theorem.[20] We do not demonstrate that the values of $\xi$ at which we wish to approximate our function do not lie on these cuts, but resort to a qualitative heuristic: Poles and zeroes of the Padé approximant that are not poles/zeroes of the function will, with increasing order of the approximant, fill out the branch cuts of the function. We plot the locations of these poles and zeroes and observe that the values of $\xi$ at which we resum lies away from the putative cuts.

The discussion so far has taken place with a fixed $y_0$. But given any pair of charge ratios $(\Lambda, \Upsilon)$, we note that there is a corresponding $y_0$ and the resulting series (97) is analytic at the origin, with some nonzero radius of convergence. This provides us with a way of solving the

---

[18]Some experimentation shows, in our examples, that the diagonal sequences give the best numerical agreement with the conjectured result.

[19]We follow the textbook [21], wherein this theorem appears with number 6.6.8.

[20]Theorem 6.6.9 of [21].

attractor/orthogonality equations for any value of the charge ratios. In practice we are limited by the number of terms in the series that we can compute, so we only ever approximate the true value with finite order Padé approximants (rather than evaluating the limit of the infinite sequence of Padé approximants).

## 4.2 Series identities

Below we display the (conjectural) identities of the type (90) we obtain for the families of Calabi–Yau manifolds discussed in [11], [14], and in this paper. Apart from AESZ4 and AESZ290, all of these sums as written diverge, and thus the identities should be regarded as concerning the analytic continuation discussed above.

**Convergent identities**

Table 1: Table of identities with convergent series.

| AESZ 4, from Ref. [11] | | | | | | | |
|---|---|---|---|---|---|---|---|
| $\Lambda$ | $\Upsilon$ | Modular form | Verified accuracy | $Y_{111}$ | $c_2$ | $Y_{110}$ | Attractor point |
| 12 | $-5$ | **54.4.a.c** | 130 figures | 9 | 54 | 1/2 | $z = -2^{-3}3^{-6}$ |

$$\frac{3\pi}{2}\frac{L(1)}{L(2)} = \sqrt{69} - \sqrt{\frac{2}{\pi^3}} \sum_{\substack{j\in\mathbf{Z}_{>0}\\ \mathfrak{p}\in\mathrm{pt}(j)}} (-1)^j \widetilde{N}_{\mathfrak{p}} \left(\frac{j}{3\pi\sqrt{69}}\right)^{l(\mathfrak{p})-1/2} K_{l(\mathfrak{p})-1/2}\left(\frac{\pi j\sqrt{69}}{3}\right).$$

| AESZ 290 | | | | | | | |
|---|---|---|---|---|---|---|---|
| $\Lambda$ | $\Upsilon$ | Modular form | Verified accuracy | $Y_{111}$ | $c_2$ | $Y_{110}$ | Attractor point |
| $\frac{c_2}{24} - \frac{5}{4}$ | $-Y_{011}$ | **14.4.a.b** | 68 figures | $-\frac{30}{13}$ | — | — | $x = 3^{-6}$ |

$$\frac{14}{3\pi}\frac{L(2)}{L(1)} = \sqrt{\frac{13}{3}} - \sqrt{\frac{13}{15\pi^3}} \sum_{\substack{j\in\mathbf{Z}_{>0}\\ \mathfrak{p}\in\mathrm{pt}(j)}} \widetilde{N}_{\mathfrak{p}} \left(\sqrt{\frac{13}{3}}\frac{j}{10\pi}\right)^{l(\mathfrak{p})-1/2} (-1)^{l(\mathfrak{p})} K_{l(\mathfrak{p})-1/2}\left(\pi j\sqrt{\frac{13}{3}}\right).$$

**Resummed identities**

Table 2: Table of identities where a divergent series must be resummed.

| AESZ 11, from Ref. [11] | | | | | | | |
|---|---|---|---|---|---|---|---|
| $\Lambda$ | $\Upsilon$ | Modular form | Verified accuracy | $Y_{111}$ | $c_2$ | $Y_{110}$ | Attractor point |
| 6 | $-3$ | **180.4.a.e** | 68 figures | 6 | 48 | 0 | $z = -2^{-4}3^{-3}$ |

$$\frac{2\pi}{5}\frac{L(1)}{L(2)} = \sqrt{39} - \sqrt{\frac{3}{\pi^3}} \sum_{\substack{j\in\mathbf{Z}_{>0}\\ \mathfrak{p}\in\mathrm{pt}(j)}} (-1)^j \widetilde{N}_{\mathfrak{p}} \left(\frac{j}{2\pi\sqrt{39}}\right)^{l(\mathfrak{p})-1/2} K_{l(\mathfrak{p})-1/2}\left(\pi j\sqrt{\frac{13}{3}}\right).$$

| **AESZ 34, from Ref. [14]** | | | | | | | |
|:---:|:---:|:---:|:---:|:---:|:---:|:---:|:---:|
| $\Lambda$ | $\Upsilon$ | Modular form | Verified accuracy | $Y_{111}$ | $c_2$ | $Y_{110}$ | Attractor point |
| 6 | $-12$ | **14.4.a.a** | 44 figures | 24 | 24 | 0 | $\varphi = -1/7$ |

$$\frac{15\pi}{7}\frac{L(1)}{L(2)} = 2\sqrt{6} - \sqrt{\frac{3}{\pi^3}}\sum_{\substack{j\in\mathbf{Z}_{>0}\\ \mathfrak{p}\in\mathrm{pt}(j)}}(-1)^j\widetilde{N}_{\mathfrak{p}}\left(\frac{j}{8\pi\sqrt{6}}\right)^{l(\mathfrak{p})-1/2}K_{l(\mathfrak{p})-1/2}\left(\pi j\sqrt{\frac{2}{3}}\right).$$

| **AESZ 22** | | | | | | | |
|:---:|:---:|:---:|:---:|:---:|:---:|:---:|:---:|
| $\Lambda$ | $\Upsilon$ | Modular form | Verified accuracy | $Y_{111}$ | $c_2$ | $Y_{110}$ | Attractor point |
| 5 | $-13$ | **825.4.a.f** | 11 figures | 35 | 50 | 1/2 | $z = -1$ |

$$-\mathrm{i}\,\frac{1+\dfrac{\mathrm{i}\pi}{33}\dfrac{L(1)}{L(2)}}{1-\dfrac{2\mathrm{i}\pi}{165}\dfrac{L(1)}{L(2)}} = \frac{\sqrt{69}}{6} - \sqrt{\frac{7}{10\pi^3}}\sum_{\substack{j\in\mathbf{Z}_{>0}\\ \mathfrak{p}\in\mathrm{pt}(j)}}\mathrm{e}^{\frac{5\pi\mathrm{i}}{7}j}\widetilde{N}_{\mathfrak{p}}\left(\frac{3j}{5\pi\sqrt{69}}\right)^{l(\mathfrak{p})-1/2}K_{l(\mathfrak{p})-1/2}\left(\frac{j\pi\sqrt{69}}{21}\right).$$

| **AESZ 118** | | | | | | | |
|:---:|:---:|:---:|:---:|:---:|:---:|:---:|:---:|
| $\Lambda$ | $\Upsilon$ | Modular form | Verified accuracy | $Y_{111}$ | $c_2$ | $Y_{110}$ | Attractor point |
| 5 | $-5$ | **825.4.a.f** | 36 figures | 10 | 40 | 0 | $z = -1$ |

$$\frac{4\pi}{165}\frac{L(1)}{L(2)} = \sqrt{\frac{5}{3}} - \frac{1}{\sqrt{5\pi^3}}\sum_{\substack{j\in\mathbf{Z}_{>0}\\ \mathfrak{p}\in\mathrm{pt}(j)}}(-1)^j\widetilde{N}_{\mathfrak{p}}\left(\frac{3j}{10\pi\sqrt{15}}\right)^{l(\mathfrak{p})-1/2}K_{l(\mathfrak{p})-1/2}\left(\pi j\sqrt{\frac{5}{3}}\right).$$

| **AESZ 17** | | | | | | | |
|:---:|:---:|:---:|:---:|:---:|:---:|:---:|:---:|
| $\Lambda$ | $\Upsilon$ | Modular form | Verified accuracy | $Y_{111}$ | $c_2$ | $Y_{110}$ | Attractor point |
| 6 | $-15$ | **14.4.a.b** | 11 figures | 30 | 36 | 0 | $z = -1$ |

$$\frac{3\pi}{14}\frac{L(1)}{L(2)} = \frac{1}{\sqrt{5}} - \frac{1}{\sqrt{15\pi^3}}\sum_{\substack{j\in\mathbf{Z}_{>0}\\ \mathfrak{p}\in\mathrm{pt}(j)}}(-1)^j\widetilde{N}_{\mathfrak{p}}\left(\frac{j}{6\pi\sqrt{5}}\right)^{l(\mathfrak{p})-1/2}K_{l(\mathfrak{p})-1/2}\left(\frac{\pi j}{\sqrt{5}}\right).$$

| **AESZ 34, from Ref. [14]** | | | | | | | |
|:---:|:---:|:---:|:---:|:---:|:---:|:---:|:---:|
| $\Lambda$ | $\Upsilon$ | Modular form | Verified accuracy | $Y_{111}$ | $c_2$ | $Y_{110}$ | Attractor point |
| 66/17 | $-192/17$ | **34.4.b.a** | 6 figures | 24 | 24 | 0 | $\varphi = 33 + 8\sqrt{17}$ |

$$-2\mathrm{i}\,\frac{204 + 31(1+2\mathrm{i})\pi\frac{L(1)}{L(2)}}{119 - 6(1+2\mathrm{i})\pi\frac{L(1)}{L(2)}}$$

$$= \sqrt{\frac{65}{3}} - \frac{17}{2\sqrt{3\pi^3}}\sum_{\substack{j\in\mathbf{Z}_{>0}\\ \mathfrak{p}\in\mathrm{pt}(j)}}\mathrm{e}^{\frac{16\pi\mathrm{i}}{17}j}\widetilde{N}_{\mathfrak{p}}\left(\frac{17j}{8\pi\sqrt{195}}\right)^{l(\mathfrak{p})-1/2}K_{l(\mathfrak{p})-1/2}\left(\frac{\pi j}{17}\sqrt{\frac{65}{3}}\right).$$

# 5 Numerical methods applied to the series

In this section we discuss checking numerically the relations conjectured in the previous section. We explain in detail the numerical methods used to obtain an approximate analytic continuation to test the identities in each special case.

For each example, we compute the first 76 terms of the series (90). Since the $j$'th term involves operations on the set of partitions of $j$, computing many more terms is not computationally feasible. The number of partitions of 76 is 9,289,091 and the growth of $p(n)$ is given by the Hardy–Ramanujan formula

$$p(n) \sim \frac{1}{4n\sqrt{3}} \exp\left( \pi\sqrt{\frac{2n}{3}} \right). \tag{98}$$

We then form the Padé approximants of these truncated sums. We are then faced with the problem of extracting as much numerical agreement as possible from the finite number of terms in (42) that we are able to compute. This is a classical problem which has been nicely addressed in [22]. In practice, we find the best numerical agreement by using diagonal Padé approximants. Moreover, by applying case-specific conformal maps

$$\xi = \xi(Z), \tag{99}$$

we can expand (97) as a power series in $Z$. The particular map $\xi(Z)$ is chosen to distance the point at which we wish to evaluate from the poles of the function $t$. The Padé approximants in the variable $Z$ (instead of $\xi$) return a yet more accurate resummation. Precisely which $\xi(Z)$ is needed to accomplish this differs drastically according to how many branch cuts $t(\xi)$ has. We do this for the examples having one or two branch cuts, but do not apply this method in the cases that have more branch cuts as we do not know a good map.

We do not have a fully reliable means of establishing the presence of these branch cuts, so we instead resort to heuristics. It is known [21,82] that the poles of the Padé approximant fill out arcs in the complex plane corresponding to the branch cuts of the original function. As the order of the Padé approximant is increased, these poles develop an accumulation point at the branch point of the function. So we guess the locations of branch cuts by forming sequences of Padé approximants and observing the accumulation of poles along arcs/lines, reading off numerical estimates for branch point locations.

In the rest of this section, we discuss each case appearing in the list of §4.2 in some detail, in particular specifying the accuracy to which the numerical evaluation of the analytic continuation agrees with the conjectured result. The degree of agreement that we find varies, with the extremes being 6 and 130 figures. We can offer some justification for the difference in accuracy across our examples, based on the position of $\xi$ relative to poles of $t$ and the number of branch cuts that the function $t(\xi)$ possesses. For completeness we also list, in each case, the rank-two attractor in the coordinates used by the form of the operator as it appears in [65,66], the topological data, charge ratios $(\Lambda_*, \Upsilon_*)$ and the expression for $t$ in terms of L-values.

We also present figures showing the poles and zeros of the approximants. This serves to illustrate how we locate branch points, but also gives some qualitative explanation as to why our final accuracy differs so drastically across examples: it is out of our hands whether the points at which we wish to resum our series are close to or far from poles of the function we are approximating, or what the number of cuts is. Very roughly, we observe the expected behaviour whereby our approximations are worse when the branch cut structure is more complicated, and poles are closer to the point where the series are evaluated.

## 5.1 Convergent examples

### AESZ4

This is the Picard–Fuchs operator for the mirror of the bicubic intersection $\mathbf{P}^5[3,3]$, shown in [11] to possess a rank-two attractor at $z = -2^{-3}3^{-6}$. For this example,

$$Y_{111} = 9, \quad c_2 = 54, \quad Y_{011} = \frac{1}{2}, \quad \chi = -144, \quad \Lambda = 12, \quad \Upsilon = -5,$$

$$t = \frac{1}{2} + \frac{\pi i}{4} \frac{L_{\mathbf{54.4.a.c}}(1)}{L_{\mathbf{54.4.a.c}}(2)}. \tag{100}$$

By summing 76 terms of the series, with find agreement to 91 figures. 15 Shanks transformations [83] improve this to 130 figures.

### AESZ290

We have explained already that there is a rank-two attractor at $x = 3^{-6}$. As for AESZ17, we must have $\chi = -30$. We cannot provide values for $c_2$ or $Y_{011}$, but by studying (40) we determine that a solution can be obtained with

$$\Lambda - \frac{c_2}{24} = -\frac{5}{4}, \qquad \Upsilon + Y_{011} = 0, \tag{101}$$

Here we are strictly only discussing a solution of (40) and we have not computed a meaningful value for the quantities $c_2$, $Y_{011}$ in terms of geometry. One could act on both sides of (40) by the real change of basis matrix that transforms the integral symplectic basis to the rational basis so that $c_2$ and $Y_{011}$ do not appear in the equation: solving this equation and transforming back to the integral basis gives the charge values in (101).

These combinations are sufficient for us to obtain a number from (42), with

$$Y_{111} = -\frac{30}{13}, \qquad t = \frac{7i}{3\pi} \frac{L_{\mathbf{14.4.a.b}}(2)}{L_{\mathbf{14.4.a.b}}(1)}. \tag{102}$$

We do not know a good geometric interpretation for the invariants $N_k^{GW}$ that are used for this sum, but they can be computed in the same way as usual.

Summing 76 terms of this series gives agreement to 61 figures, and two Shanks transformations improve this to 68 figures.

## 5.2 Resummed examples

From 76 terms we form a diagonal Padé approximant of order 38. I.e. the numerator and denominator are both degree 38 polynomials. We find no advantage in using nondiagonal approximants.

### AESZ11

This is the Picard–Fuchs operator for the mirror of the $\mathbf{WP}^5_{(1^4,2)}[4,3]$ intersection, shown in [11] to have a rank-two attractor point at $z = -2^{-4}3^{-3}$. With the data

$$Y_{111} = 6, \quad c_2 = 48, \quad Y_{011} = 0, \quad \chi = -156, \quad \Lambda = 6, \quad \Upsilon = -3,$$

$$t = \frac{1}{2} + \frac{\pi i}{15} \frac{L_{\mathbf{180.4.a.e}}(1)}{L_{\mathbf{180.4.a.e}}(2)}, \tag{103}$$

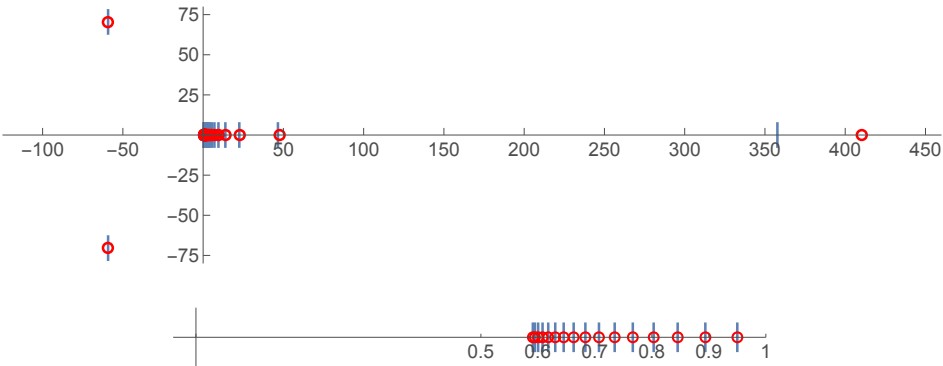

Figure 3: AESZ11. Red circles show poles of the order 38 diagonal Padé approximant, while blue lines give zeroes. Our first plot displays every pole/zero, with the second plot zooming in to show an accumulation point. We interpret the line of poles as giving a branch cut.

we get a formal identity from (42), with a divergent sum. Our Padé resummed value at $\xi = -1$ gives agreement to 48 figures. We give a plot that shows the locations of zeroes and poles of the Padé approximant in the complex plane in Figure 3. Based on the Figure, we believe that for this example the function that we are approximating has a single branch cut on the real axis. For this order 38 approximant, the pole closest to the origin (an approximation of the branch point from which the cut emanates) is located at

$$p = 0.59113388184858\ldots \tag{104}$$

This decimal expansion gives the location of a pole in our approximant, but we do not claim that this is a good approximation to the true location of the branch cut (which higher order approximants would more accurately reproduce). By applying the conformal map

$$\xi(Z) = -\frac{4pZ}{(1-Z)^2}, \tag{105}$$

and forming a new order 38 diagonal Padé approximant in $Z$, we obtain 65 figures of agreement. Instead of using the above algebraic map, we also try the uniformising map

$$\xi(Z) = p\left(1 - e^Z\right), \tag{106}$$

and from this obtain 68 figures of agreement.

### AESZ34, $\varphi = -1/7$

This rank-two attractor was found in [14]. A divergent sum can be formed from (42) using the data

$$Y_{111} = 24, \quad c_2 = 24, \quad Y_{011} = 0, \quad \chi = -16, \quad \Lambda = 6, \quad \Upsilon = -12,$$

$$t = \frac{1}{2} + \frac{5\pi i}{28} \frac{L_{14.4.a.a}(1)}{L_{14.4.a.a}(2)}. \tag{107}$$

Our Padé resummation gives 44 figures of agreement.

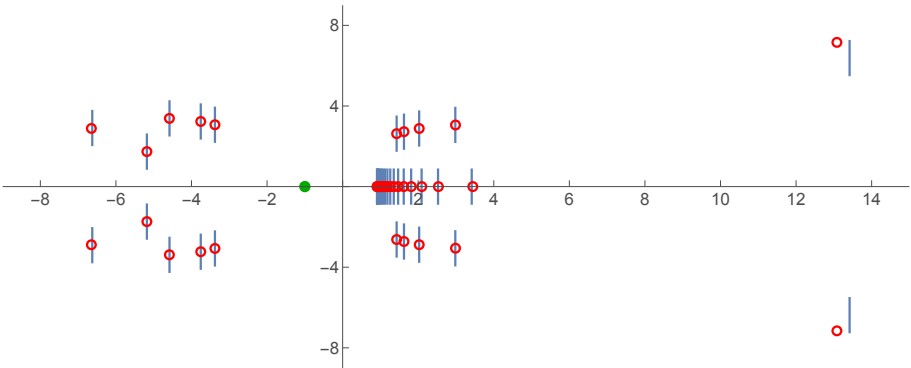

Figure 4: AESZ34, $\varphi = -1/7$. Poles and zeroes of the order 38 diagonal Padé approximant for this example. The green dot at $\xi = -1$ indicates the point to which we are trying to analytically continue our series. It is unclear from this plot what the branch cut structure is, but there would appear to be at least three cuts.

For this example we are unable to come up with a helpful conformal mapping, as we did for AESZ11. While for AESZ11 there was one "obvious" branch cut (and useful conformal maps that help in approximating functions with single poles are known), this is not so clear for this series with our available data.

### AESZ22

We have found an attractor at $z = -1$. Inserting the data

$$Y_{111} = 35, \quad c_2 = 50, \quad Y_{011} = \frac{1}{2}, \quad \chi = -50, \quad \Lambda = 5, \quad \Upsilon = -13,$$

$$t = \frac{1}{2}\left(1 - \frac{2\pi i}{165}\frac{L_{\mathbf{825.4.a.f}}(1)}{L_{\mathbf{825.4.a.f}}(2)}\right)^{-1}, \tag{108}$$

into (42) produces a divergent series. We find agreement to 7 figures, this being another example that only weakly supports our conjectured expressions.

Based on Figure 5, we expect branch cuts extending from infinity to the points

$$
\begin{aligned}
p_1 &= -0.30494498943495\ldots, \\
p_2 &= \phantom{-}0.1402133297157\ldots
\end{aligned}
\tag{109}
$$

A conformal map well suited to there being two branch points is [22]

$$\xi(Z) = \frac{-4p_1p_2Z}{-p_1(1+Z)^2 + p_2(1-Z)^2}. \tag{110}$$

Applying the Padé approximant after using this map leads to 11 figures of agreement. Reference [22] also discusses the utility of a two-cut uniformizing map, but in this example of ours this map only gave improvement to 10 figures.

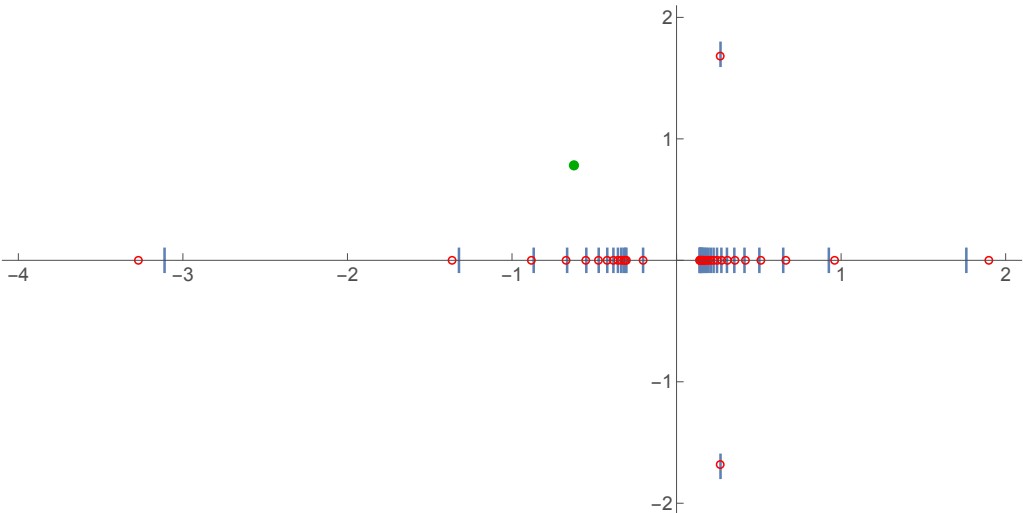

Figure 5: AESZ22. We display all poles and zeroes of this example's order 38 diagonal Padé approximant, with a green dot at $\xi = \exp\left(\frac{5\pi i}{7}\right)$ where we wish to continue our series. This plot strongly suggests a pair of branch cuts on the real axis.

### AESZ118

Here we provide a rank-two attractor at $x = -2^{-5}$, which we have explained is the same point as AESZ22's rank-two attractor at $z = -1$. To construct a sum, we use the data

$$Y_{111} = 10, \quad c_2 = 40, \quad Y_{011} = 0, \quad \chi = -50, \quad \Lambda = 5, \quad \Upsilon = -5,$$

$$t = \frac{1}{2} + \frac{2\pi i}{165} \frac{L_{\mathbf{825.4.a.f}}(1)}{L_{\mathbf{825.4.a.f}}(2)}. \tag{111}$$

Note that this sum uses a *different* set of Gromov–Witten invariants as compared to the AESZ22 example. It may be of interest that the enumerative invariants of derived equivalent, non-birational geometries are related to the same L-values by our computations. Our Padé resummation gives 29 figures of agreement.

The sequence of diagonal Padé approximants indicates the presence of singularities at

$$\begin{aligned} p_1 &= -6.97656721804019\ldots, \\ p_2 &= \phantom{-}0.21843573615225\ldots \end{aligned} \tag{112}$$

These are poles of the order 38 approximant. Our plot makes a good indication that a branch cut starts at $p_2$ and extends to infinity. It is tempting to speculate that another branch point starts at approximately $p_1$ and extends to negative infinity, and that the plot would make this more apparent at higher orders. Whether or not this is a branch point or a pole of our function is immaterial for the use of conformal maps to improve resummation accuracy. The map (110) in this case leads to 36 figures of agreement, while the two-cut uniformization map of [22] leads to 35 figures.

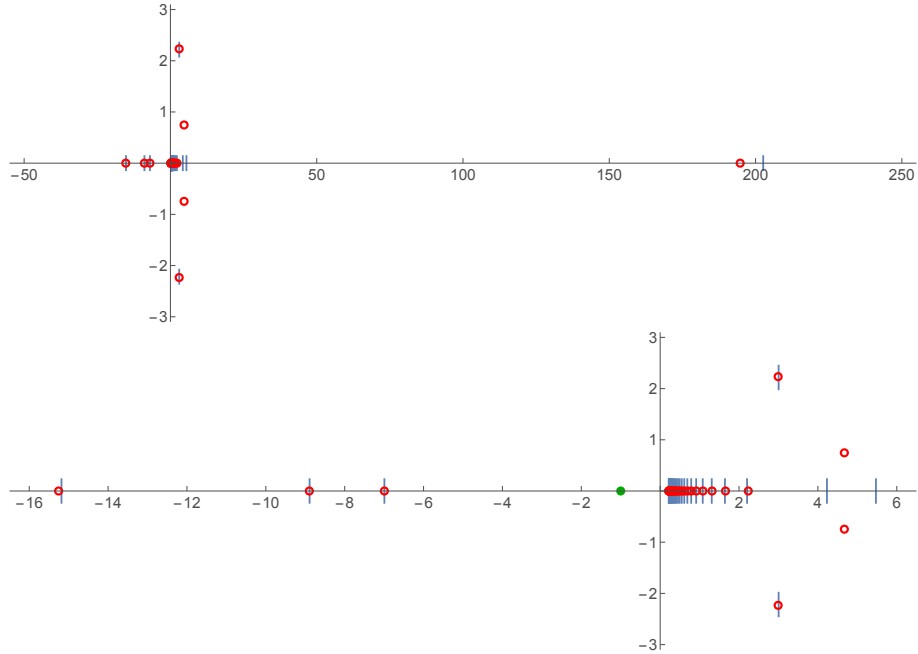

Figure 6: AESZ118. The first plot shows all poles and zeroes of the order 37 diagonal Padé approximant for the AESZ118 series. The second plot zooms in to our region of interest. The green dot at $\xi = -1$ is where we want to continue our sum to. For the order 38 approximant, a spurious pole-zero pair (with residue $\approx 10^{-27}$) appears for small negative $\xi$, which we do not consider to truly reflect any analytic property of the function.

### AESZ17

We have provided a rank-two attractor at $z = -1$, and explained that this is the same as AESZ290's. Once again, the sum that we arrive at uses a different set of Gromov–Witten invariants. The necessary data is

$$Y_{111} = 30, \quad c_2 = 36, \quad Y_{011} = 0, \quad \chi = -30, \quad \Lambda = 6, \quad \Upsilon = -15,$$

$$t = \frac{1}{2} + \frac{3\pi i}{28} \frac{L_{\mathbf{14.4.a.b}}(1)}{L_{\mathbf{14.4.a.b}}(2)}. \tag{113}$$

Here our numerics give agreement to 11 figures.

The plot showing poles and zeroes for these examples makes a strong case that there are three branch cuts. We do not know a good conformal map for such a configuration, and so do not have an improvement to report beyond the 11 figures mentioned.

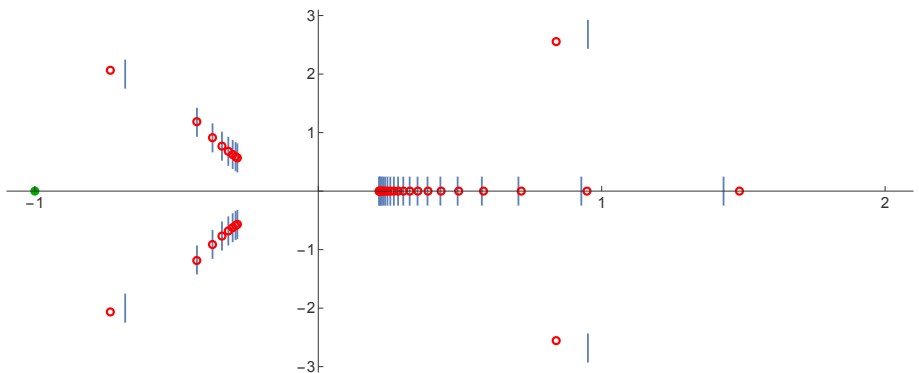

Figure 7: AESZ17. Poles and zeroes of the order 38 diagonal Padé approximant, with the green dot at $\xi = -1$ indicating where we want to continue our function to.

**AESZ34,** $\varphi = 33 + 8\sqrt{17}$

This is another rank-two attractor found in [14]. Unlike the other examples that we consider in this paper this attractor is not rational, instead belonging to a quadratic extension of **Q**. The same was true of the example discussed in [1]. Some details of the point counting problem for such varieties are discussed in [14], from which we take the conjectural L-value evaluations of the periods. A divergent sum can be formed from (42) using the data

$$Y_{111} = 24, \quad c_2 = 24, \quad Y_{011} = 0, \quad \chi = -16, \quad \Lambda = \frac{66}{17}, \quad \Upsilon = -\frac{192}{17},$$

(114)

$$t = \frac{1}{6} + 1156 \left( 2856 - 45\pi i \left(9 + \sqrt{17}\right) \frac{L_{\mathbf{34.4.b.a}}(1)}{L_{\mathbf{34.4.b.a}}(2)} \right)^{-1}.$$

This is our worst-supported example, as we only find 6 figures of agreement. Just as for the other AESZ34 series, we are unable to come up with a helpful conformal mapping for this example.

**A comment on accuracy**

Across our examples, there is a striking difference in final accuracies. We have better accuracy when there are fewer branch cuts, and when the point at which we evaluate (the green square

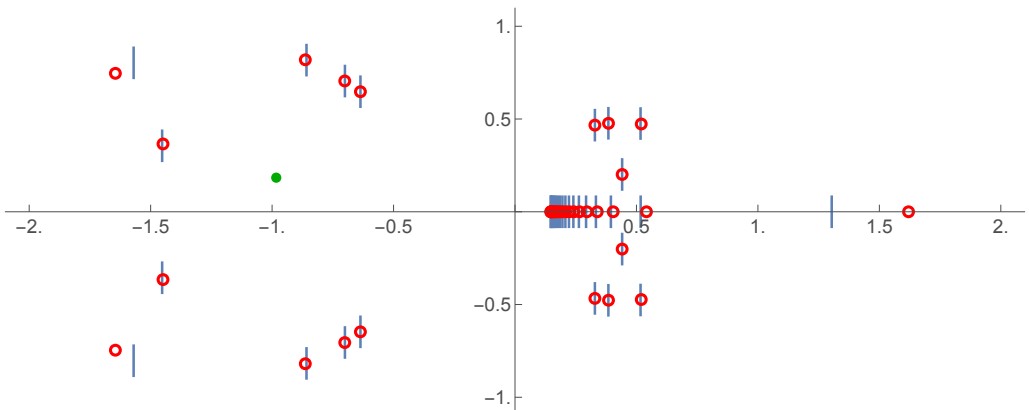

Figure 8: AESZ34, $\varphi = 33 + 8\sqrt{17}$. Poles and zeroes for this example's order 38 diagonal Padé approximant. We have placed a green dot at $\xi = \exp\left(\frac{16\pi i}{17}\right)$, where we want to evaluate our series. We cannot easily recognise the number of branch cuts.

of our plots) is further from the nearest pole. Both of these issues are known to slow down convergence of Padé approximants [21]. Based on this consideration, we still present our weakest supported examples and do not conclude that the very low number of figures in those cases is due to the claimed identity being incorrect. It may well be the case that a further complicating factor that we have neglected (for instance in the case of AESZ22) corrects some of our summation identities, but based on the examples that we have verified to a higher accuracy it seems that any such additional effect is not universal. We suggest that the weak examples are weak because of very slow convergence rather than an incorrect assumption, as the construction of each sum follows the same procedure of studying supergravity, identifying a modular variety, and applying the mirror map.

## Acknowledgments

We thank Pyry Kuusela for collaboration in the early stages of this work, and for a large number of conversations on the matters of this paper and related issues. We are very glad to be able to acknowledge helpful conversations with Jean-Emile Bourgine, Luca Cassia, Albrecht Klemm, Alan Lauder, Robert Pryor, and Jay Swar. We are grateful to Duco van Straten for extensive discussions on periods, modular forms, and twists. We thank Kilian Bönisch for some timely and helpful advice on L-values and monodromies, and Mohamed Elmi for sharing some useful code and discussions. JM thanks Emanuel Scheidegger for interesting discussions on quasiperiods and AESZ22 inter alia, and the Beijing International Centre for Mathematical Research for generous hospitality during the final stages of this work. JM thanks Johanna Knapp for collaborative discussions during ongoing work to further study AESZ17. We thank Erik Panzer for kind instruction on Padé resummation and other discussions. JM thanks the organisers of the very interesting *Elliptics and Beyond '23* workshop for the opportunity to discuss this work, and for their gracious hospitality in Zürich. JM also thanks the organisers of the *New Deformations of Quantum Field and Gravity Theories* scientific program hosted by the Matrix Institute, where this work was finished. We also thank the organisers of *The Geometry of Moduli Spaces in String Theory* scientific program hosted by the Matrix Institute, where this work was finished again.

**Funding information** JM was supported by EPSRC studentship #2272658, with his research now supported by a University of Melbourne Establishment Grant. For the purpose of open access, the authors have applied a CC-BY public copyright license to any Author Accepted Manuscript (AAM) version arising from this submission.

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
