# Peer review of "Classical Weight-Four L-value Ratios as Sums of Calabi--Yau Invariants"

_SciPost Physics, doi:SciPost Phys. 18, 181 (2025)_

## Round 2 · Referee Report · Anonymous (Referee 1) · 2025-3-19

Strengths

  1. The paper reports on (numerical) evidence of a conjectural remarkable identity between L-function values on the one hand and suitably evaluated Mellin transforms of modular forms on the other hand, which in enumerative geometry corresponds to a relationship between GW invariants and point counts over finite fields of Calabi-Yau manifolds.

  2. The paper reports on newly discovered attractor points of in families of Calabi-Yau threefolds.

  3. To carry out the numerical checks of the conjectured identities and to reach higher numerical precision, the authors use Padé's approximation as an innovative tool in this context. This technique may prove useful in similar contexts in the future as well.

Weaknesses

  1. From a conceptual point of view, the paper does not offer deep new insights. Nevertheless, the results can be useful for future investigations as any new example is relevant in this demanding research field.

  2. As the authors state themselves, there are some weaknesses in the precession of the numerical evidence.

Report

I believe that the submitted manuscript is an interesting research article, which offers solid and important results. While there are some numerical weaknesses the paper offers convincing numerical evidence for the above mentioned conjectured number theoretic identity. Therefore, I recommend publication of the paper as it stands.

Requested changes

No requested changes.

Recommendation

Publish (easily meets expectations and criteria for this Journal; among top 50%)

---

## Round 2 · Referee Report · Anonymous (Referee 2) · 2025-5-4

Report

This paper continues the progress, in part due to previous work by the authors of this paper, on rank-2 attractors. This refers to points in Calabi-Yau moduli space which satisfy the black hole attractor equations for two independent sets of charges. While relatively rare, finite lists of examples have been found in recent years, and any additions to these lists are useful additions to the overall picture of such Calabi-Yaus. Almost always, such points have very interesting number-theoretical properties such as the ones discussed in this paper about L-function values.

In the paper two new rank-2 attractors are found and discussed. Further, the remarkable conjectural number theoretical identity 2.41 is verified numerically to large orders. This identity relates L-values to Gromov-Witten invariants. One nice addition in this paper is the use of Pade approximants to extend the range of such formulas that can be studied.

There is a short but sufficient Section 2 with the introductory concepts, and the rest of the paper is well-written and readable. I recommend the paper for publication as it stands.

Recommendation

Publish (easily meets expectations and criteria for this Journal; among top 50%)

---

## Editorial Decision

published